# JoyAgents-R1: Accelerating Multi-Agent Evolution Dynamics with Variance-Reduction Group Relative Policy Optimization

## Abstract

Large Language Model (LLM)-based multi-agent systems represent a promising paradigm with broad applicability, exemplified by general-purpose Artificial Intelligence (AI) assistants capable of performing multiple tasks. Nevertheless, joint optimization across functionally distinct agents remains challenging due to divergent working modes and reward functions. To address this issue, we introduce JoyAgents-R1, a framework that accelerates multi-agent evolution with a novel Variance-Reduction Group Relative Policy Optimization (VR-GRPO), integrating efficient sampling and update strategies. Specifically, VR-GRPO performs Monte Carlo sampling based on an initial reasoning trajectory to avoid the exponential explosion of the joint action space while maintaining policy diversity. Then, the method selects the top-$K$ sampling groups with maximal reward fluctuations based on the marginal benefit principle, thereby enabling cost-effective parameter updates. To further complement evolution, an adaptive memory evolution mechanism that repurposes GRPO rewards as cost-free supervisory signals is designed to eliminate repetitive reasoning and accelerate convergence. Experiments on multi-task AI assistant datasets across both general and e-commerce scenarios demonstrate that JoyAgents-R1, built upon smaller 3B/7B open-source models, achieves performance comparable to that of larger LLMs, such as DeepSeek-R1, and surpasses DeepSeek-V3 by an average of 6%.

## 1 Introduction

The rapid advancement of LLMs Achiam et al. (2023); Anthropic (2024); Grattafiori et al. (2024); Bai et al. (2023); Yang et al. (2024) has revolutionized agent-based systems, empowering agents with the ability to perform reasoning, planning, and natural language interaction across diverse domains Guo et al. (2024); Gao et al. (2025). Compared to single agents with specialized functionalities Li et al. (2023); Ruan et al. (2023); Qin et al. (2024); Dong et al. (2024), multi-agent systems Wan et al. (2025); Liao et al. (2025); Dang et al. (2025) demonstrate superior flexibility and scalability in tackling complicated tasks such as general-purpose AI assistants Fu et al. (2024) and emergency responder management Sivagnanam et al. (2024), among which Multi-Agent Reinforcement Learning (MARL) is a key technique of the community Ning & Xie (2024).

Since Reinforcement Learning (RL) has demonstrated remarkable efficacy in aligning models with human preferences Ouyang et al. (2022), LLM-based MARL methods have flourished and achieved certain results in sophisticated task decomposition Iqbal et al. (2022); Tian et al. (2023) and adaptive coordination Fu et al. (2022); Li et al. (2024). In MARL, the behavior of one agent may affect the rewards of other agents, which may cause environmental instability and lead to low system efficiency and performance Hernandez-Leal et al. (2017). While methods like Multi-Agent Proximal Policy Optimization (MAPPO) Yu et al. (2022) have advanced MARL by adapting PPO Schulman et al. (2017) to multi-agent settings, their reliance on additional value functions introduces critical limitations. Moreover, the decoupling of policy and value updates in actor-critic architecture often leads to training instability, particularly when coordinating heterogeneous agents with functionally distinct roles and potentially misaligned reward structures Zhong et al. (2024b), which poses severe challenges to the dynamics of multi-agent evolution.

(a) Naive sampling leads to an exponential explosion

(b) Monte Carlo sampling based on an initial reasoning trajectory

Figure 1: Different sampling strategies for multi-agents. Green circles indicate nodes sampling multiple actions via GRPO, while gray circles involve no additional sampling. Compared to naive sampling in (a), (b) could avoid the exponential explosion of the multi-agent joint action space.

Recently, DeepSeek-R1 Guo et al. (2025) introduced Group Relative Policy Optimization (GRPO) Shao et al. (2024), a novel RL framework that improves LLM decision-making by replacing critic models with population-based comparisons. GRPO generates multiple responses per input and selects actions based on relative group advantages, significantly reducing computational cost. While GRPO has shown strong performance in single or homogeneous agents Deng et al. (2024); Xia & Luo (2025); Lu et al. (2025) and vision-language models Huang et al. (2025), directly applying it to multi-agent systems remains challenging. As shown in Fig. 1 (a), the joint action space grows exponentially with the number of agents and creates far more complex action spaces than those in single-agent scenarios Hernandez-Leal et al. (2020); Liu et al. (2024b), requiring tailored sampling and update strategies. Furthermore, updating all agents with diverse architectures and dynamic reasoning paths simultaneously is also challenging and remains unsolved.

To address the above issues, we introduce JoyAgents-R1, a novel framework that accelerates multi-agent evolution through Variance-Reduction Group Relative Policy Optimization (VR-GRPO). To the best of our knowledge, this is the first work to apply GRPO in heterogeneous multi-agent joint evolution, which will offer novel insights to the community. JoyAgents-R1 integrates efficient sampling, update strategies, and adaptive memory evolution to enable faster convergence and improved system performance. As shown in Fig. 1 (b), VR-GRPO utilizes Monte Carlo sampling based on an initial reasoning trajectory, thereby avoiding the exponential explosion of the joint action space while maintaining policy diversity. In addition, we follow the marginal benefit principle Kauder (2015) to update agents associated with the top-$K$ sampling groups exhibiting the largest variance of intra-group rewards, maximizing joint utility at minimal computational cost. To facilitate multi-agent training, we design a memory evolution mechanism harnessed from GRPO rewards as a "free lunch" based on the insight that these rewards are inherently coupled to memory. Through direct utilization of GRPO rewards to update reasoning-associated memory, the decision-making and memory modules achieve synchronous optimization, effectively alleviating the difficulty of joint evolution.

To sum up, the main contributions of this work can be listed as follows:

- We introduce JoyAgents-R1, a novel joint evolution framework for multi-agent systems. To the best of our knowledge, this is the first work to adapt GRPO for functionally distinct multi-agents, enabling synergistic enhancement of their decision-making and memory capabilities.

- We propose VR-GRPO tailored for multi-agent training. It integrates a new Monte Carlo sampling strategy to efficiently navigate the joint action space, and a selection strategy to guide joint GRPO updates. With the marginal benefit principle, it maximizes the joint utility with minimal computational overhead.

- We design a simple yet effective memory mechanism derived from GRPO rewards to facilitate the multi-agent evolution. During LLM parameter updates, agents' memories evolve synchronously based on action rewards, significantly accelerating training and boosting reasoning performance.

- We conduct extensive experiments on multi-task AI assistant benchmarks, including both general and domain-specific scenarios. JoyAgents-R1, built upon smaller open-source models, demonstrates performance comparable to or exceeding that of significantly larger state-of-the-art LLMs, such as DeepSeek-R1 and DeepSeek-V3.

Due to length limitations, the full review of related work is provided in Section A.1.

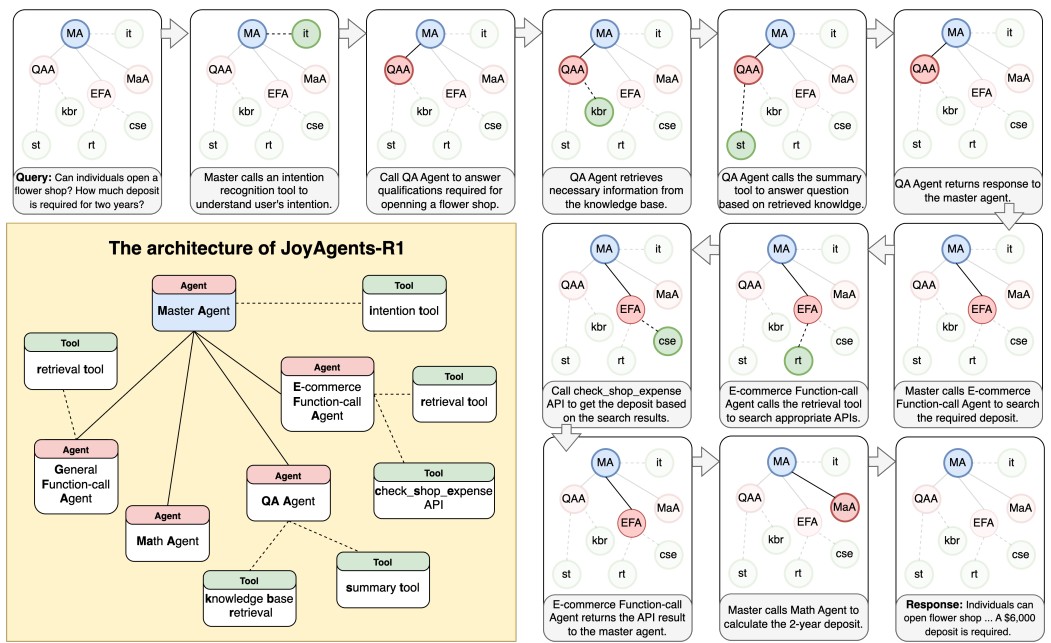

Figure 2: The multi-agent architecture and a reasoning example of JoyAgents-R1.

## 2 METHOD

We introduce JoyAgents-R1, a novel joint evolution dynamics for multi-agent reinforcement learning. First, a hierarchical architecture is designed to integrate functionally distinct multi-agents for collaborative tasks (Section 2.1). Then, a variance-reduction GRPO including efficient Monte Carlo sampling and marginal benefit-driven updating is constructed for joint training (Section 2.2). Finally, an adaptive memory evolution mechanism leveraging GRPO rewards as cost-free supervisory signals is proposed for synchronized optimization of agent decision modules (Section 2.3).

### 2.1 THE ARCHITECTURE OF JOYAGENTS-R1

As illustrated in Fig. 2, the proposed JoyAgents-R1 adopts a hierarchical architecture, consisting of a master agent and multiple sub-agents as follows:

- **Master agent** first analyzes the query, then orchestrates sub-agents (*e.g.*, question answering) or tools (*e.g.*, intention recognition) in each step, and determines the final response to the user.

- **Question-answering agent** performs general and domain-specific (*e.g.*, e-commerce) question answering by retrieving and summarizing the recalled information from external knowledge bases.

- **Function-call agents** include general-purpose and domain-specific (*e.g.*, e-commerce) agents. These agents either execute the function call directly via memory-driven prompts or invoke a tool retriever and make further selections from the recalled APIs.

- **Math agent** is specialized for solving mathematical problems.

Fig. 2 shows an example of a reasoning chain during inference. Upon receiving a query, the master agent analyzes the user intent and assigns tasks to specialized sub-agents, which execute iterative operations using contextual inputs transmitted by the master until completion. Then, the results are relayed back to the master for subsequent planning cycles until termination. Each agent executes in a ReAct manner Yao et al. (2023) and dynamically retrieves validated strategies from memory to minimize redundant reasoning and enhance decision efficiency. On the other hand, since the inherent complexity of actor-critic frameworks with critic models, JoyAgents-R1 employs GRPO for policy optimization during training, which foregoes the value function and computes advantages in a group-relative manner Shao et al. (2024). However, its direct application in multi-agent joint evolution faces the following challenges:

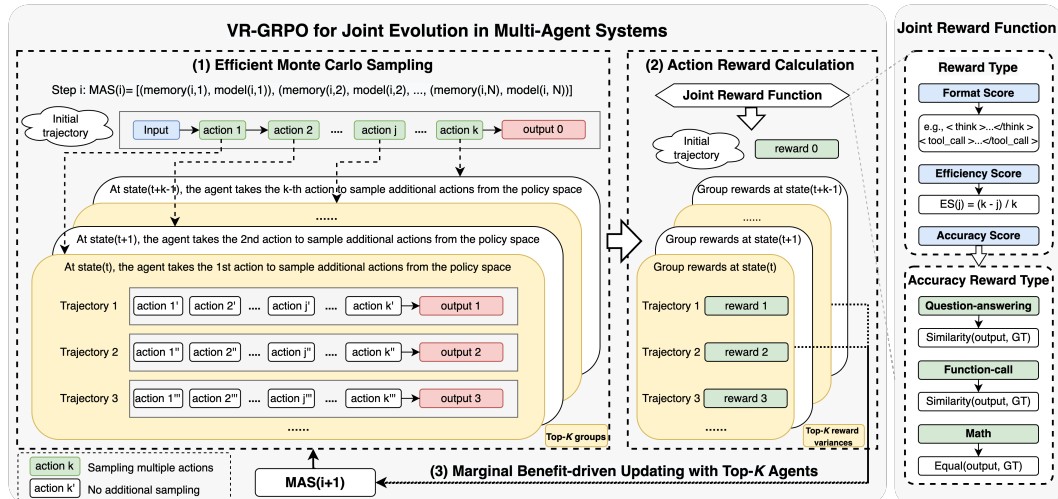

Figure 3: VR-GRPO for joint evolution in the Multi-Agent Systems (MAS) comprises three steps: (1) Efficient Monte Carlo sampling to sequentially sample agent actions along an initial trajectory, mitigating action space explosion. (2) Action reward calculation, incorporating accuracy, format, and efficiency rewards based on trajectory output. (3) Marginal benefit-driven updating, which prioritizes the top-$K$ agents with the highest reward variance to accelerate policy evolution.

- **Low sampling efficiency.** Since multiple actions are sampled for each agent throughout the reasoning chain, the number of trajectories explodes exponentially as shown in Fig. 1.
- **Inefficient parameter updates.** Sequential policy updates for all agents in the chain will lead to inefficient parameter optimization. Furthermore, decoupled policy adjustments struggle to coordinate inter-agent dependencies for overall performance enhancement.
- **Slow training convergence.** Diverse architectures and outputs hinder policy synchronization, making functionally distinct multi-agents prone to convergence difficulties during training.

## 2.2 VARIANCE-REDUCTION GROUP RELATIVE POLICY OPTIMIZATION

To address the above challenges, we propose Variance-Reduction Group Relative Policy Optimization (VR-GRPO), tailored for multi-agent systems (Fig. 3), including the following parts:

**Efficient Monte Carlo sampling.** For a multi-agent system (MAS) consisting of $N$ agents $\{m_i\}_{i=1}^N$, each agent $m_i$ has an action space including $G_i$ possible actions. Therefore, when GRPO is directly applied to MAS in a naive way, a reasoning trajectory of length $k$ will generate $G_{Naive} = G_1 \times G_2 \cdots \times G_k$ sampling paths in an exponential explosion (Fig. 1 (a)). Different from that, VR-GRPO performs efficient Monte Carlo sampling as shown in Fig. 3 (1). Specifically, given a query $q$ obtained from the dataset, VR-GRPO first produces an initial trajectory of length $k$, and then sequentially samples $(G_i - 1)$ actions for each node $m_i$ in the trajectory. To facilitate comparative calculations and address the exponential explosion problem, the reasoning path from the query to the sampling node (*e.g.*, $m_j$) remains along the original trajectory. After $m_j$, no more sampling is conducted, and reasoning continues until the end. In this way, a total of $G_{NMC} = G_1 + G_2 \cdots + G_k$ trajectories are generated. This addition operation is much smaller than the multiplication one, which effectively improves the sampling efficiency, especially for MAS with long trajectories.

**Marginal benefit-driven updating.** For the $k$ groups of sampled trajectories, the model parameters of corresponding agents $\{m_i\}_{i=1}^k$ are $\Theta = \{\theta_i\}_{i=1}^k$. Given a query-answer pair $(q, a)$, the old policy $\pi_{\theta_i old}$ of each agent $m_i$ samples a group of outputs $\{o_j\}_{j=1}^{G_i}$. Then, the policy model $\pi_{\theta_i}$ is optimized by maximizing the objective as follows:

$$\mathcal{J}_{GRPO}(\theta_i) = \mathbb{E}[q \sim P(Q), \{o_j\}_{j=1}^{G_i} \sim \pi_{\theta_i old}(O|q)]$$

$$\frac{1}{G_i} \sum_{j=1}^{G_i} \left( \min \left( \frac{\pi_{\theta_i}(o_j|q)}{\pi_{\theta_i old}(o_j|q)} A_j, \text{clip} \left( \frac{\pi_{\theta_i}(o_j|q)}{\pi_{\theta_i old}(o_j|q)}, 1 - \epsilon, 1 + \epsilon \right) A_j \right) \right) \quad (1)$$

---

**Algorithm 1** Dynamic Memory Updating from GRPO Rewards

---

**Input**: Planning chain $\mathcal{P}$, Trajectory output $\mathcal{O}$, Trajectory reward $\mathcal{R}_{\mathcal{M}}$, Upper bound $U$, Lower bound $L$,
      Set of $n$ recalled memories $Recall_q$ given query $q$, Hyperparameters $\alpha$ and $\beta$, Timestamp $t$

---

1: **for** each memory $\mathcal{M}_i \in Recall_q$ **do**
2:     Compute similarity: $sim_i \leftarrow Sim(\mathcal{O}, \mathcal{O}_i)$ (Direct answer mode) **or** $Sim(\mathcal{P}, \mathcal{P}_i)$ (Tool call mode)
3:     **if** $\mathcal{R}_{\mathcal{M}} > U$ **then**
4:         Update timestamp: $t_i \leftarrow t$;
5:         Compute time and reward differences: $\Delta t \leftarrow -|t - t_i|$ **and** $\Delta s \leftarrow s_i \cdot |\mathcal{R}_{\mathcal{M}} - U|$
6:     **else if** $\mathcal{R}_{\mathcal{M}} < L$ **then**
7:         Compute time and reward differences: $\Delta t \leftarrow -|t - t_i|$ **and** $\Delta s \leftarrow -s_i \cdot |\mathcal{R}_{\mathcal{M}} - L|$;
8:         Update timestamp: $t_i \leftarrow t$
9:     **end if**
10:    Update recalled memory scores: $\mathcal{R}_{\mathcal{M}i} \leftarrow \mathcal{R}_{\mathcal{M}i} + \alpha\Delta t + \beta\Delta s$
11: **end for**
12: **for** each memory $\mathcal{M}_i \notin Recall_q$ **do**
13:    Compute time difference: $\Delta t \leftarrow -|t - t_i|$;
14:    Update timestamp: $t_i \leftarrow t$;
15:    Update recalled memory scores: $\mathcal{R}_{\mathcal{M}i} \leftarrow \mathcal{R}_{\mathcal{M}i} + \alpha\Delta t$
16: **end for**

---

where $\epsilon$ is a hyper-parameter for the clipped objective. $A_j$ denotes the advantage of the $j$-th response by normalizing the group-wise rewards with the average and standard deviation. To foster the dynamic strategy adaptation crucial for multi-agent coordination, the standard KL penalty is eliminated for greater policy divergence, without the computational overhead of maintaining multiple reference models. Therefore, the objective for updating all models straightforwardly is as follows:

$$\mathcal{J}_{Multi-GRPO}(\Theta) = \{\mathcal{J}_{GRPO}(\theta_i) \mid i = 1, \cdots, k\} \tag{2}$$

Furthermore, to accelerate the policy update and enable the perception of global states, VR-GRPO implements a variance-reduction objective based on the marginal benefit principle Kauder (2015):

$$\mathcal{J}_{VR-GRPO}(\Theta) = \{\mathcal{J}_{GRPO}(\theta_i) \mid i \in \text{argtopK}\left(\text{Var}(R_i), K\right)\} \tag{3}$$

where $R_i = \{r_j\}_{j=1}^{G_i}$ is the reward set obtained after sampling $G_i$ actions from $m_i$ to the end of trajectory. $\text{argtopK}(\cdot)$ returns the index set of the first $K$ nodes with the largest reward variance. To this end, VR-GRPO prioritizes updating model parameters for the top-$K$ agents exhibiting the largest performance fluctuations among all reasoning trajectory participants. Compared to updating all agents sequentially as shown in Equation 2, this variance-aware selection strategy minimizes computational overhead while maximizing the joint benefit, efficiently steering multi-agent parameter updates through GRPO rewards in a dynamic paradigm.

**Action rewards.** As shown in Fig. 3 (2), given the final answer $a$ corresponding to $q$ as the ground truth, the reward $\mathcal{R}$ (*i.e.*, $r_i$ in the above text) of each agent action consists of three terms as follows:

$$\mathcal{R} = \mathcal{R}_{\mathcal{A}} + \mathcal{R}_{\mathcal{F}} - \mathcal{R}_{\mathcal{E}} \tag{4}$$

- *Accuracy reward ($\mathcal{R}_{\mathcal{A}}$).* The accuracy reward is calculated end-to-end from the final trajectory answer. The metrics are tailored for tasks' distinct settings and output formats. For instance, semantic similarity assesses alignment with ground truth for question answering and function calling, while mathematical operations require exact-match validation against predefined solutions.

- *Format reward ($\mathcal{R}_{\mathcal{F}}$).* Since each agent infers in the RaAct style, the model output is formatted using HTML tags for thinking (*i.e.*, $< think > \cdots < /think >$ ) and tool calling (*i.e.*, $< tool\_call > \cdots < /tool\_call >$ ). Format rewards could guide the model to generate structured results, improving clarity and enhancing the reasoning ability of LLMs.

- *Efficiency reward ($\mathcal{R}_{\mathcal{E}}$).* For the $j$-th node in a trajectory of length $k$, its efficiency score is computed as $\mathcal{R}_{\mathcal{E}} = \frac{k-j}{k}$, which imposes a penalty proportional to the distance of the node from the trajectory endpoint, where the efficiency score is quantified by number of subsequent decision steps required, showing how the current node's plan impacts downstream computational cost.

### 2.3 FREE LUNCH IN GRPO REWARDS FOR MEMORY EVOLUTION

For functionally distinct multi-agents, memory modules are introduced to accelerate model training and reduce redundant reasoning. To improve the efficiency of joint evolution, the agent memory is designed to undergo dynamic adaptation alongside LLMs' updates during training. Different from previous methods that require training dedicated models or utilizing LLMs to evolve memory, JoyAgents-R1 creates a simple yet effective memory updating mechanism that leverages GRPO rewards as a cost-free supervisory signal and mainly consists of the following three steps:

**Adaptive reward thresholding.** The memory of each agent along a trajectory is updated using a unified reward without efficiency score (*i.e.*, $\mathcal{R}_\mathcal{M} = \mathcal{R}_\mathcal{A} + \mathcal{R}_\mathcal{F}$), which ensures consistent update criteria and enables independent memory modules to perceive the overall performance. For $G_{NMC}$ trajectories sampled from query $q$, the mean $\mu$ and standard deviation $\sigma$ of corresponding rewards are first computed, then the 2.5% and 97.5% percentiles of the approximate normal distribution are selected as the lower ($L = \mu - 1.96\sigma$) and upper ($U = \mu + 1.96\sigma$) bounds for memory updates.

**Dynamic memory updating.** The algorithm first determines whether to add a new memory based on the upper bound $U$. Then, it dynamically updates the memories recalled by the query $q$. For other memories, they are updated only according to time decay as shown in Algorithm 1.

**Memory overflow handling.** To ensure memory quality and save storage space, the memory will be deleted either when its final reward ($\mathcal{R}_{\mathcal{M}i}$) falls below a predefined threshold $D$ or when the memory capacity exceeds upper bounds and the memory's reward rank is relatively low.

To this end, the memory module is synergistically updated with model parameters through trajectory rewards, accelerating training convergence and boosting reasoning performance.

## 3 EXPERIMENTS

### 3.1 IMPLEMENTATION DETAILS

In this work, we opt for the Qwen2.5 series model Yang et al. (2024) as the agent backbone to ensure technical reproducibility. The experiment consists of two main stages. In the first stage, the base models are fine-tuned with a learning rate of 5e-6 for 5 epochs. In the second stage, the multi-agent system is trained via reinforcement learning at a learning rate of 1e-6 for 5 epochs. Specifically, each agent from an initial trajectory is sampled $G_i = 5$ actions with a temperature of $1.2$. Subsequently, the top-5 nodes are selected for model updates. Similar to DeepSeek-R1 Guo et al. (2025), the iterative RL with GRPO is executed for 2 iterations. Regarding memory evolution, the deletion threshold is set to $D = 0$, with hyperparameters $\alpha = \beta = 1$. The models are trained on 8 NVIDIA H200 GPUs, and the best results are reported. More details are provided in the Appendix A.2.

### 3.2 DATASETS AND SETUP

To verify the effectiveness of the proposed method, we construct a multi-task dataset for multi-agent AI assistant scenarios, including general and e-commerce fields, as follows:

**Supervised fine-tuning datasets.** The input integrates diverse elements, such as user queries, retrieved memory, optional tools, historical dialogues, and tool-generated responses. The target comprises the reasoning processes, tool calling, or final answers. The master agent, responsible for dynamic reasoning and orchestrating four sub-agents, is trained on 13,000 samples: 10,000 individual sub-agent calls (2,500 per agent) and 3,000 collaborative calls. The QA agent uses 1,000 samples, including 700 real-world e-commerce cases and 300 open-domain instances from COIG Zhang et al. (2023). Regarding function-calling, the e-commerce agent is trained on 12 common e-commerce APIs with 2,000 samples, while the general-purpose agent uses 1,000 API calls from ToolBench Qin et al. (2024), totaling 3,500 instances. For the math agent, we use GSM8K Cobbe et al. (2021). More dataset configurations are provided in the Appendix A.5.

**Reinforcement learning datasets.** The RL dataset contains query-response pairs with ground-truth annotations. The training set includes 100 samples per sub-agent task (*i.e.*, math, e-commerce/general function calls, QA) and 200 collaborative instances. The test set has 500 samples, including 100 instances for each independent task and 100 cases for the collaborative task.

Table 1: Accuracies (%) of multi-tasks with methods based on larger closed- or open-source models. 'EFC' and 'GFC' are e-commerce and general function calls. Under the open-source setting, bold is the optimal and '‗' is the suboptimal value. Closed-source methods (∗) are only used for reference.

| Model | Math | QA | EFC | GFC | Cooperation | Average |
|---|---|---|---|---|---|---|
| Claude3.5-sonnet ∗ | 98.0 | 38.0 | 1.0 | 62.5 | 2.0 | 40.3 |
| GPT-4o mini ∗ | 47.0 | 31.0 | 0.0 | 65.5 | 0.0 | 28.7 |
| GPT-4o ∗ | 85.0 | 35.0 | 56.0 | 83.0 | 6.0 | 53.0 |
| DeepSeek-R1 | **98.0** | 37.0 | 24.0 | 72.0 | 7.0 | **47.6** |
| DeepSeek-V3 | 96.0 | 32.0 | 10.0 | 68.0 | 2.0 | 41.6 |
| Qwen2.5-32B | 72.0 | **38.0** | 3.0 | 72.0 | 2.0 | 37.4 |
| Qwen2.5-14B | 80.0 | 29.0 | 0.0 | 42.0 | 1.0 | 30.4 |
| Multiagent Debate (14B with 3 roles) | 95.0 | 33.0 | 0.0 | 40.6 | 1.0 | 33.9 |
| Qwen2.5-3B (Single agent with RL) | 56.0 | 9.0 | 0.0 | 58.5 | 1.0 | 24.9 |
| Qwen2.5-7B (Single agent with RL) | 81.0 | 11.0 | 0.0 | 67.2 | 1.0 | 32.0 |
| Qwen2.5-14B (Single agent with RL) | 82.0 | 27.0 | 2.0 | 63.6 | 1.0 | 35.1 |
| JoyAgents-SFT (7B Master + 3B Sub agents) | 65.0 | 13.0 | 42.0 | 73.0 | 3.0 | 39.2 |
| **JoyAgents-R1** (3B Master + 3B Sub agents) | 68.0 | 22.0 | **48.0** | **76.0** | 6.0 | 44.0 |
| **JoyAgents-R1** (7B Master + 3B Sub agents) | 75.0 | 35.0 | 46.0 | 73.1 | **9.0** | **47.6** |

## 3.3 COMPARATIVE EXPERIMENTS

**Comparison rules.** Comparisons on the multi-task benchmark and more evaluations on the publicly available ToolBench dataset are conducted. Detailed settings are provided in the Appendix A.3.

**Comparisons on the multi-task benchmark.** Multiple comparisons demonstrate the effectiveness of our approach: (1) While state-of-the-art closed-source models like GPT-4o exhibit the highest overall performance, our JoyAgents-R1 (7B Master + 3B Sub agents) achieves a competitive average accuracy of 47.6%, matching DeepSeek-R1 and significantly surpassing DeepSeek-V3 (41.6%). Notably, our method excels in the specialized E-commerce Function Call (EFC) task, achieving scores up to 48.0%, which are substantially higher than those of DeepSeek-R1 (24.0%) and DeepSeek-V3 (10.0%). This highlights the effectiveness of our domain-specific optimization on a compact architecture, which can outperform larger generalist models in targeted domains. All prompts used here are presented in the Appendix A.6. (2) When compared with open-source models, JoyAgents-R1 consistently surpasses the entire Qwen2.5 series on the average score, including the larger 32B variant, underscoring its superior parameter efficiency. Due to the limited size of our math and QA agents (3B), competing with larger models is challenging. However, through joint optimization, our approach (75% and 35%) significantly outperforms the SFT trained 7B+3B JoyAgents variant (65% and 13%). (3) We also compare with Multiagent Debate Du et al. (2024), which is implemented based on Qwen2.5-14B with three roles. Even so, our method (47.6%) significantly outperforms its performance (33.9%), further showing the efficiency and effectiveness of our proposed framework. (4) In addition, we trained individual Qwen2.5-3B/7B/14B models using reinforcement learning. For most metrics, performance improved with increasing model scale, with the exception of GFC, where the 14B model slightly underperformed the 7B variant. Nevertheless, the fine-tuned 14B model still surpassed the original 14B baseline. Under RL settings with similar model parameters, our JoyAgents-R1 (3B Master + 4 ∗ 3B Subagents, 44.0%) is significantly better than a single Qwen2.5-14B with RL (35.1%) on average, verifying the superiority of multi-agent reinforcement learning. (5) Finally, our 7B master-base method overall outperforms the 3B one, demonstrating the approach's scalability. Increasing only the master agent's size improves summarization and planning, but does not guarantee performance improvements for all sub-agents.

**More comparisons on the Toolbench dataset.** Table 2 shows consistent results across in-domain and out-of-domain settings. Under in-domain, JoyAgents-R1 outperforms both DeepSeek-R1 and DeepSeek-V3 in Plan_ACC, Act_EM, hard_F1, F1, and Average scores with notable improvements. For instance, its average score (51.2%) is much larger than DeepSeek-R1's (47.1%) and DeepSeek-V3's (43.1%). Remarkably, JoyAgents-R1 achieves these results with just 3B parameters, while DeepSeek models use over 37B parameters. Compared to larger Qwen2.5 models, JoyAgents-R1 demonstrates that smaller models can achieve competitive performance through multi-agent training. Additionally, it surpasses GPT-4o in Hard_F1 and F1 scores, both in-domain (31.5% and 39.0%) and out-of-domain (37.0% and 48.9%). In comparison with baseline methods, enabling the think process

Table 2: Comparative results (%) on the Toolbench dataset. Bold represents the optimal score, and '_' represents the suboptimal value. 'JoyAgents-SFT-no' means training without a thinking process.

| Method | Plan_ACC | Act_EM | Easy_F1 | Hard_F1 | F1 | No_Hallu | Avg |
|---|---|---|---|---|---|---|---|
| **In the domain** | | | | | | | |
| GPT-4o | **82.5** | **57.0** | **31.1** | 22.2 | _37.3_ | _99.9_ | **55.0** |
| DeepSeek-R1 | 72.3 | 45.2 | _18.7_ | 18.2 | 28.5 | **100.0** | 47.1 |
| DeepSeek-V3 | 68.4 | 38.0 | 14.9 | 14.3 | 23.6 | 99.4 | 43.1 |
| Qwen2.5-32B | 72.7 | 43.7 | 15.1 | 21.0 | 28.8 | 99.3 | 46.8 |
| Qwen2.5-14B | 42.2 | 15.0 | 3.7 | 7.1 | 10.1 | **100.0** | 29.7 |
| JoyAgents-SFT-no (7B+3B) | 55.7 | 33.0 | 10.1 | 20.5 | 26.5 | 95.7 | 40.3 |
| JoyAgents-SFT (7B+3B) | 73.3 | 47.0 | 16.3 | _29.6_ | 36.8 | 94.5 | 49.6 |
| JoyAgents-R1 (3B+3B) | _73.5_ | _48.5_ | 17.6 | **31.5** | **39.0** | 96.7 | _51.2_ |
| **Out of the domain** | | | | | | | |
| GPT-4o | **83.9** | **63.2** | **42.9** | 22.3 | 44.1 | 99.8 | **59.3** |
| DeepSeek-R1 | 74.6 | 52.0 | 28.3 | 20.4 | 35.6 | 99.8 | 51.8 |
| DeepSeek-V3 | _77.2_ | 56.1 | 32.0 | 21.9 | 38.2 | **100.0** | 54.2 |
| Qwen2.5-32B | 28.6 | 0.0 | 0.0 | 0.0 | 0.0 | **100.0** | 21.4 |
| Qwen2.5-14B | 43.9 | 16.6 | 7.2 | 6.2 | 12.3 | _99.9_ | 31.0 |
| JoyAgents-SFT-no (7B+3B) | 59.7 | 41.1 | 19.7 | 25.9 | 36.5 | 97.3 | 46.7 |
| JoyAgents-SFT (7B+3B) | 72.3 | 57.0 | _32.3_ | _36.1_ | _48.7_ | 96.3 | 57.1 |
| JoyAgents-R1 (3B+3B) | 73.4 | _57.5_ | 31.1 | **37.0** | **48.9** | 96.9 | _57.5_ |

Table 3: Ablation study for multi-agent reasoning on multiple tasks. From left to right, whether to train with RL (*vs.* SFT), whether to generate the think process, whether to update top-$K$ models (*vs.* update all models), whether to use efficiency rewards, and whether to integrate memory modules. 'Update Steps' corresponds to the best-performing model. 'Rounds' are reasoning rounds.

| Method | RL | Think | Top-$K$ | Efficiency | Memory | Update Steps↓ | Accuracy↑ | Rounds↓ |
|---|---|---|---|---|---|---|---|---|
| $M_1$ | ✗ | ✗ | - | - | - | - | 17.2 | **5.4** |
| $M_2$ | ✗ | ✔ | - | - | - | - | 35.0 | 7.1 |
| $M_3$ | ✔ | ✔ | ✗ | ✔ | ✔ | 1380 | 40.0 | 7.4 |
| $M_4$ | ✔ | ✔ | ✔ | ✗ | ✔ | 1750 | 42.4 | 8.1 |
| $M_5$ | ✔ | ✔ | ✔ | ✔ | ✗ | 2464 | 40.0 | 7.7 |
| $M_6$ | ✔ | ✔ | ✔ | ✔ | ✔ | **1112** | **44.0** | 7.8 |

in JoyAgents-SFT-no(think)[1] improves the average score from 40.3% to 49.6%, with a further 3.2% relative gain through multi-agent reinforcement learning. These results highlight the framework's strengths in advancing agent planning and API calling through coordinated policy adaptation.

## 3.4 ABLATION STUDY

Table 3 illustrates the ablation results based on Qwen2.5-3B models for multi-agents as follows:

**Effectiveness of generating the think process.** $M_1$ separately trains each agent's base model via SFT, yielding the lowest accuracy (17.2%). Compared to $M_1$, $M_2$ further incorporates think process generation and significantly improves accuracy to 35.0%, demonstrating the indispensable role of explicit reasoning in agent decision-making.

**Effectiveness of training with reinforcement learning.** Different from $M_2$, $M_6$ integrates GRPO for the joint evolution of multi-agents and enhances the accuracy by 25.7% relatively. Furthermore, the results from $M_3$ to $M_6$ using RL are much better than $M_2$, indicating the effectiveness of global perception and preference alignment in joint training with reinforcement learning.

---

[1]Unless specified otherwise, the experimental results for JoyAgents-SFT are based on implementations that integrate training with the thinking process by default.

Table 4: Ablation results (%) in updating top-$K$ models. 'Avg' is the average score.

| Method | Math | QA | EFC | GFC | Coop | Avg |
|---|---|---|---|---|---|---|
| Top-1 | 67.0 | 16.0 | **48.0** | 75.0 | 4.0 | 42.0 |
| Top-2 | 57.0 | 12.0 | 41.0 | 75.0 | **6.0** | 38.2 |
| Top-All | 64.0 | 17.0 | 44.0 | 72.0 | 3.0 | 40.0 |
| Top-5 | **68.0** | **22.0** | 48.0 | **76.0** | **6.0** | **44.0** |

Table 5: Ablation results (%) on the number of sub-agents. 'Coop' is the cooperation task.

| Method | QA | EFC | Math | Coop |
|---|---|---|---|---|
| Master + 2 sub | **26.0** | **55.0** | - | - |
| Master + 3 sub | 25.0 | 51.0 | **69.0** | 5.0 |
| Master + 4 sub | 22.0 | 48.0 | 68.0 | **6.0** |

**Effectiveness of utilizing efficiency rewards.** Since $M_4$ excludes $\mathcal{R}_\mathcal{E}$ in Eq. 4 during training, it exhibits suboptimal accuracy (42.4%) and a larger number of training steps (1750), with a maximum reasoning round of 8.1, demonstrating that efficiency constraints are critical for balancing performance and computational cost. In contrast, $M_1$ consumes the fewest inference rounds as insufficient reasoning capacity, which forces premature termination from ineffective decision-making.

**Effectiveness of updating top-$K$ models.** $M_3$ sequentially updates all models (Eq. 2), while $M_6$ improves accuracy by 10% relatively through targeted updates of top-$K$ nodes with maximal reward fluctuations and requires the least number of update steps (1112). Inspired by marginal benefit, $M_6$ prioritizes models requiring the most optimization to achieve the maximum benefit with the minimum cost, thereby improving performance and accelerating training convergence. Table 4 shows that the optimal top-k is 5, with further analysis in Appendix A.4.

**Ablation on the number of sub-agents.** Table 5 compares different numbers of sub-agents. For non-collaborative tasks, fewer sub-agents are more efficient due to reduced complexity and interference. For collaborative tasks, the full set of 4 sub-agents outperforms the others, benefiting from a broader range of data and interactions, highlighting the effectiveness of our multi-agent system design. More discussion is available in the Appendix A.4.

**Effectiveness of integrating memory modules.** $M_5$ disables memories across all agents, relying solely on decision modules. This leads to a relative decrease of 10.0% in accuracy compared to $M_6$ and a maximum update steps (2464), highlighting the crucial role of memory modules. In Fig. 4, JoyAgents-R1 with memory exhibits greater stability and reaches peak performance faster during training. The method using memory reaches its peak at step 140, while the one without memory reaches at step 168, showing that our memory module enhances convergence speed. More case analyses are provided in the Appendix A.7.

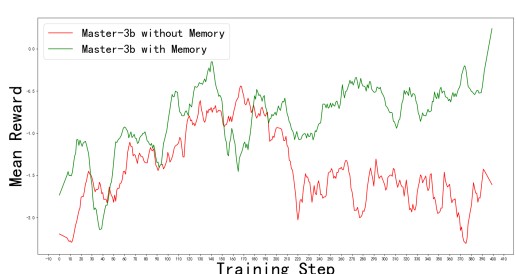

Figure 4: JoyAgents-R1 with memory achieves higher reward peaks faster during training.

## 4 CONCLUSION AND DISCUSSION

**Conclusion.** This paper introduces JoyAgents-R1, a framework that accelerates multi-agent evolution dynamics through VR-GRPO. By integrating initial trajectory-based Monte Carlo sampling with marginal benefit-driven evolution strategies, we enhance sampling efficiency and training convergency across functionally distinct multi-agents. The adaptive memory evolution mechanism leveraging GRPO rewards further reduces redundant reasoning and accelerates convergence. Comparative experiments demonstrate that JoyAgents-R1, built on smaller open-source models, surpasses DeepSeek-V3 by 6%. Extensive ablation studies confirm that VR-GRPO and memory modules effectively accelerate model training and improve performance.

**Limitations.** Due to computational constraints, the proposed method currently focuses on multi-agent systems based on small open-source models. In addition, existing training frameworks lack compatibility with heterogeneous multi-agent joint training. Future work will focus on scaling up LLMs to achieve performance gains and cross-domain robustness, while engineering a dedicated framework optimized for heterogeneous multi-agent co-evolution with computational efficiency.

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

# A   APPENDIX

This supplementary material details the proposed method and presents additional experimental results. Section A.1 is related work. Section A.2 presents more implementation details for experiments. Section A.3 reports extra comparative results, and Section A.4 introduces more analysis for ablation studies. Section A.5 describes dataset configurations. Section A.6 includes all prompts used in baselines and our multi-agent architecture. Section A.7 provides extended case analyses. Finally, Section A.8 introduces the usage of LLMs. The code is included in *JoyAgents_R1_Code.zip*.

## A.1   RELATED WORK

### A.1.1   LLM-BASED MULTI-AGENT PLANNING

Recent breakthroughs in LLMs have transformed the landscape of agent planning Huang et al. (2024). Autonomous agents can implement iterative self-reflection mechanisms, dynamically integrate external information via structured prompts Yao et al. (2023); Shinn et al. (2023); Zhou et al. (2024); Lin et al. (2023); Qiao et al. (2024), and perceive environments to plan tasks through sophisticated reasoning and decision-making processes Ghallab et al. (2004). Compared with single-agent approaches that struggle with inefficiency and environmental adaptability, multi-agent systems achieve robust performance through decentralized decision-making and collaborative mechanisms, enabling the coordination of agents with distinct capabilities and objectives to pursue shared goals in fields like robotics Kannan et al. (2024), tool calling Shen et al. (2024), and AI assistants Fu et al. (2024). However, few multi-agent systems can achieve multi-domain tasks. Moreover, open-source LLMs Grattafiori et al. (2024); Yang et al. (2024) lag significantly behind state-of-the-art models, which are either closed-source with opaque mechanisms Achiam et al. (2023); Anthropic (2024) or overly complex for multi-agent deployment Guo et al. (2025). Recent work has modeled agents as computational graphs, such as GPTSwarm Zhuge et al. (2024), which optimizes orchestration via prompt refinement and graph connectivity, while AFLOW Zhang et al. (2024b) automates workflow generation with Monte Carlo Tree Search. In addition, Multiagent Debate Du et al. (2024) and its variants prompt multiple LLMs to iterate the debate to improve reasoning. This work introduces a hierarchical multi-agent architecture to interpret user queries and perform dynamic planning. Based on the smaller open-source LLMs, our framework implements diverse capabilities, including question answering, mathematical computation, and tool calling, revealing the mechanisms that drive effective heterogeneous multi-agent collaboration in resource-constrained environments.

### A.1.2   MULTI-AGENT REINFORCEMENT LEARNING

MARL has witnessed substantial advancements, rendering it an ideal approach for tackling complex and challenging tasks Yuan et al. (2025). This work focuses on cooperative MARL tasks where various agents share a common goal, which has been successfully applied in many fields such as game playing Wang et al. (2021); Yu et al. (2022), task allocation Iqbal et al. (2022), skill discovery Yang et al. (2023), and circuit design Wang et al. (2024). Typical MARL methods employ an actor-critic framework, where actors generate actions based on observations, and critics evaluate their long-term efficacy Sun et al. (2024). There are policy-based methods like MADDPG Lowe et al. (2017) and MAPPO Yu et al. (2022) and value-based ones like VDN Sunehag et al. (2018) and QMIX Rashid et al. (2020). Although recent studies have explored LLM-based MARL frameworks for problem-solving Chen et al. (2023); Hong et al. (2024) and embodied intelligence Mandi et al. (2024); Zhang et al. (2024a); Kannan et al. (2024), these approaches primarily focus on enhancing inter-agent communication and cooperative decision-making, with limited emphasis on the joint evolution of multi-agent systems. In addition, many methods adopt parameter sharing across agents, which restricts their applicability to homogeneous scenarios Deng et al. (2024); Grammel et al. (2020); Christianos et al. (2021) and fails to address heterogeneous systems Zhong et al. (2024b). Recently, GRPO Shao et al. (2024) has eliminated the value function and relies on observed rewards, which is suitable for joint training of heterogeneous multi-agents Ke et al. (2025). Based on GRPO, MLPO Estornell et al. (2025) trains the leader in multi-agent systems, but there are few attempts to train all agents jointly. Since the dynamic and changeable reasoning paths in multi-agents, direct sampling based on GRPO will lead to an exponential explosion. Therefore, we propose VR-GRPO that combines efficient Monte Carlo sampling with marginal benefit optimization to guide reasoning path sampling and agent updating.

Table 6: Training settings for supervised fine-tuning.

| Hyperparameter | Value |
|---|---|
| learning_rate | 5e-6 |
| max_length | 16384 |
| num_train_epochs | 5 |
| max_grad_norm | 1 |
| weight_decay | 0.01 |
| warmup_ratio | 0.03 |
| lr_scheduler_type | cosine |
| optim | adamw_torch |
| gradient_accumulation_steps | 4 |
| dataloader_num_workers | 8 |
| per_device_train_batch_size | 1 |

Table 7: Training settings for reinforcement learning.

| Hyperparameter | Value |
|---|---|
| learning_rate | 1e-6 |
| max_length | 16384 |
| num_train_epochs | 5 |
| max_grad_norm | 1 |
| adam_epsilon | 1e-5 |
| num_groups | 5 |
| topk_groups | 5 |
| kl_coef | 0 |
| grpo_epoch | 2 |
| policy_clip_eps | 0.2 |
| temperature | 1.2 |
| per_device_train_batch_size | 1 |

### A.1.3 LLM-BASED AGENT MEMORY

Agent memory can be divided into RAG-based and embodied categories Huang et al. (2024). The former is typically stored in additional storage, while the latter embeds memories into model parameters by fine-tuning LLMs. In this work, we focus on RAG-based long-term memory mechanisms. Recent works have explored diverse strategies Packer et al. (2023); Lu et al. (2023); Wang et al. (2023); Huang et al. (2023). Methods such as MoT Li & Qiu (2023), TiM Liu et al. (2023), and RAP Kagaya et al. (2024) aim to improve LLM reasoning and planning by leveraging memories after selection or thinking. MemoryBank Zhong et al. (2024a) draws inspiration from the Ebbinghaus forgetting curve Ebbinghaus (2013) to design a selective information retention mechanism. HELPER Sarch et al. (2023), ExpeL Zhao et al. (2024), RET-LLM Modarressi et al. (2024), Synapse Zheng et al. (2024), and A-mem Xu et al. (2025) adopt different approaches for knowledge aggregation, storage, and retrieval, enhancing LLMs' adaptability to novel tasks. Moreover, there are several memory mechanisms tailored for multi-agent systems Zhang et al. (2024c), exploring the memory synchronization Chen et al. (2024), communication Mandi et al. (2024), and the information asymmetry Light et al. (2023) among agents. Nevertheless, existing memory modules often struggle to keep pace with LLM updates, thereby limiting system efficacy. In contrast, we present the joint evolution dynamics where agent memory and decision-making modules evolve synergistically with LLMs optimization. This mechanism leverages GRPO rewards as cost-free supervisory signals, eliminating the need for dedicated model training while enhancing convergence efficiency.

### A.2 IMPLEMENTATION DETAILS

To improve the reproducibility of the experiments, more training settings for supervised fine-tuning (SFT) and reinforcement learning (RL) are provided in Table 6 and Table 7, respectively. To compare

the SFT and RL methods fairly, the former is trained for a total of 10 epochs, while the latter is trained with SFT for 5 epochs followed by RL for 5 epochs. For RL training, the models are deployed on the TRL von Werra et al. (2020) framework on 2 NVIDIA H200 GPUs for accelerated inference via vLLM Kwon et al. (2023) and real-time weight updates, while allocating 6 GPUs for joint training.

## A.3 COMPARATIVE EXPERIMENTS

**Evaluation metrics**   The metrics are categorized into accuracy and efficiency. For accuracy, the calculations vary across task types. In question-answering tasks, accuracy is measured by the semantic similarity between the predicted and ground truth answers, with a threshold of 0.6. For mathematical problems, accuracy is binary (0 or 1) based on exact numerical matching. For function-call tasks, a response with correct function names is scored 1, otherwise 0. Regarding efficiency, it is quantified by the average number of steps required to complete a task, reflecting the reasoning efficiency of the method.

**Experimental setup for the multi-task benchmark.**   To verify the effectiveness of the proposed method, a comprehensive comparison is conducted with agents based on closed-source and open-source models. All baselines run under a consistent set of prompts, and each model is assessed as a single agent through React-based multi-turn interactions. For the proposed multi-agent method, the model parameters activated for each query typically range from 6B (master plus a single sub-agent) to a maximum of 15B (master plus 4 sub-agents).

**Experimental setup for the Multiagent Debate.**   We have implemented and compared the Multiagent Debate with our method. We opt for Qwen2.5-14B as the backbone and expand it into three with different roles. As stated on Lines 140-143 of the supplementary material, modifications are made based on the original system prompt for the single agent as follows:

- The first role is changed as: *You are a customer service expert of an e-commerce platform, specializing in answering user questions based on retrieved e-commerce knowledge.*
- The second role is changed as: *..., specializing in selecting appropriate tools to solve user problems based on user questions.*
- The third role retains its original role as: *..., specializing in selecting appropriate tools or retrieving relevant e-commerce knowledge to solve user problems.* Moreover, add the rule at the end as: *Here are two responses from agents regarding the current issue or a call to a specific tool. Please analyze the correctness of these responses and, based on your understanding, select one or generate a new action.* agent 1: {agent_1_response}, agent 2: {agent_2_response}.
- In each round of reaction, modify the original single LLM response generation to: use the first two roles to generate two responses, then pass these two responses to the third role to obtain the final response, which serves as the think and action for that round.

**Experimental setup for the Toolbench dataset.**   To validate the effectiveness of the proposed JoyAgents-R1, more comparative experiments are conducted on ToolBench Qin et al. (2024). The benchmark involves integrating API calls to accomplish tasks, where the agent should select the correct API and compose necessary API requests accurately. In this section, the test set is divided into in- and out-of-domain based on whether the tools used in the test instances have been seen during training. This setup enables us to evaluate both the learning and generalization capabilities of the method. Moreover, the proposed method is compared with two baseline approaches, namely Single-think-SFT and Single-nothink-SFT. They are based on the Qwen2.5-3B model and fine-tuned on the toolbench training set. The former outputs the reasoning process and final results, while the latter directly generates the final response without the reasoning process. We also evaluate the performance of the larger open- and closed-source LLMs Yang et al. (2024); Guo et al. (2025); Liu et al. (2024a); Achiam et al. (2023) are also used to build agents for comparisons without fine-tuning.

**More metrics on the Toolbench dataset.**   To comprehensively compare the model performance, a variety of evaluation indicators are used as follows:

- Plan_ACC: The accuracy of the agent's planning decisions at each step of the tool calling.

Table 8: Accuracies (%) of multi-tasks with agents based on larger SOTA closed-source or open-source models. 'FC' is the function call.

| Model | Math | QA | E-commerce FC | General FC | Cooperation | Average |
|---|---|---|---|---|---|---|
| GPT-4o | $84.3 \pm_{1.2}$ | $\mathbf{35.0} \pm_{2.0}$ | $\mathbf{50.3} \pm_{4.9}$ | $76.2 \pm_{0.1}$ | $5.0 \pm_{1.0}$ | $\mathbf{50.2} \pm_{1.3}$ |
| DeepSeek-R1 | $\mathbf{98.0} \pm_{1.0}$ | $34.7 \pm_{3.2}$ | $19.3 \pm_{4.2}$ | $72.4 \pm_{1.8}$ | $5.3 \pm_{3.2}$ | $45.9 \pm_{1.5}$ |
| DeepSeek-V3 | $95.7 \pm_{1.5}$ | $32.3 \pm_{5.5}$ | $6.3 \pm_{3.2}$ | $\mathbf{77.2} \pm_{1.3}$ | $3.0 \pm_{3.5}$ | $42.9 \pm_{2.3}$ |
| Qwen2.5-32B | $71.0 \pm_{1.7}$ | $34.2 \pm_{3.6}$ | $3.0 \pm_{1.0}$ | $68.9 \pm_{0.5}$ | $1.7 \pm_{0.6}$ | $35.8 \pm_{0.9}$ |
| Qwen2.5-14B | $80.0 \pm_{4.0}$ | $30.3 \pm_{1.5}$ | $0.3 \pm_{0.6}$ | $43.0 \pm_{1.1}$ | $2.0 \pm_{0.0}$ | $31.2 \pm_{1.2}$ |
| JoyAgents-SFT (7B+3B) | $65.0 \pm_{1.0}$ | $19.3 \pm_{5.7}$ | $39.3 \pm_{2.1}$ | $63.5 \pm_{1.1}$ | $5.0 \pm_{1.0}$ | $38.4 \pm_{0.9}$ |
| JoyAgents-R1 (3B+3B) | $70.0 \pm_{1.7}$ | $20.7 \pm_{2.3}$ | $48.3 \pm_{0.6}$ | $73.9 \pm_{0.7}$ | $\mathbf{7.0} \pm_{1.7}$ | $44.0 \pm_{0.8}$ |

- Act_EM: The proportion of predicted API names that exactly match the real API names.
- Easy_F1: The predicted argument F1 score when the ground truth argument is empty.
- Hard_F1: The predicted argument F1 score when the ground truth argument is not empty.
- F1: The predicted argument F1 score across all conditions.
- No_Hallu: The frequency of predicted API names that do not have hallucinations.
- Avg: The average value of the above indicators.

**More comparative results on the multi-task benchmark.** We have conducted 5 runs on Table 1 to calculate the mean and error bars as shown in Table 8. For the mean value, the proposed method still shows superior performance compared with the open source model-based methods similar to Table 1 in the main text. JoyAgents-R1 (44.0%) still has better average performance than the larger DeepSeek-V3 (42.9%) and achieves the best performance in the collaborative task. For the standard deviation, JoyAgents-R1 has the smallest fluctuation in average accuracy, with only 0.82. These results further verify the effectiveness of the proposed method.

A.4 More Analysis for Ablation study

**Ablation on updating top-$K$ models.** Table 4 presents the performance outcomes of updating varying numbers of nodes along a trajectory. The empirical results reveal that updating the top-5 nodes yields optimal performance, outperforming alternative strategies. Specifically, compared to updating only the top-1 node, the top-5 update achieves a 37.5% improvement on the QA dataset and a 50% gain in collaborative tasks. Relative to updating top-2 nodes, the top-5 approach delivers a 19.3% boost on the math dataset and an 83.3% enhancement on QA tasks. compared to updating all nodes, updating the top-5 nodes results in a 29.4% improvement on QA and a 100% increase in collaborative task success rates. These findings demonstrate that merely updating the top-1 or top-2 nodes is insufficient for holistic system optimization, as such localized adjustments fail to address systemic weaknesses. Conversely, updating all nodes lacks global awareness and focus, leading to redundant computations. Our top-5 strategy, however, employs global trajectory analysis to identify and update the weakest nodes, *i.e.*, those most limiting system performance, thereby maximizing efficiency and efficacy. This selective updating mechanism ensures that optimization efforts are concentrated on critical bottlenecks, yielding superior overall performance.

**Ablation on the number of sub-agents.** Table 5 presents the performance of the master agent when integrated with varying numbers of sub-agents across different datasets. Specifically, the configurations include:

- *2 sub-agents*: QA agent and e-commerce function-call agent.
- *3 sub-agents*: The aforementioned two plus the math agent, with the addition of collaborative tasks.

The results indicate that for individual tasks, configurations with fewer sub-agents (2 or 3) out-perform the full set of sub-agents. For instance, on the QA dataset, the 2-sub-agent setup yields an 18.2% improvement, while the 3-sub-agent setup achieves a 13.6% gain. Similarly, on the e-commerce function-call dataset, the 2-sub-agent and 3-sub-agent configurations exhibit 14.6% and

6.2% improvements, respectively. These findings align with our intuition that fewer sub-agents are more effective for non-collaborative tasks, due to reduced complexity and interference. Conversely, in collaborative tasks, the full set of 4 sub-agents demonstrates superior performance, attributed to their exposure to a broader range of data and interactions. This outcome underscores the efficacy of our multi-agent system design, which is specifically tailored to address collaborative challenges. The enhanced performance in collaborative scenarios validates the structural design of our system, highlighting the benefits of a comprehensive multi-agent framework in handling complicated and interdependent tasks.

## A.5 DATASETS AND SETUP

The datasets used in this paper can be divided into two categories: SFT and RL. Since the RL stage is trained end-to-end, this type of data only contains the initial query and the final response. More details of the SFT dataset used for each agent are introduced as follows:

**Master agent datasets.** As shown in Table 9, the case for the master agent includes user queries, optional tools, invoked agents, retrieved memories, historical dialogues, and tool-generated responses. In addition, the reasoning processes (*i.e.*, $< think > \cdots < /think >$), tool calling (*i.e.*, $< tool\_call > \cdots < /tool\_call >$), and final answers are provided in the dataset.

**Function-call agent datasets.** Regarding function-calling, the e-commerce agent incorporates 12 prevalent APIs of e-commerce platforms, while the general agent draws from 1,000 diverse function calls from ToolBench Qin et al. (2024). Table 10 illustrates a case for the e-commerce function-call agent, which is called the 'rody agent' in our multi-agent system for easy reference in code and expressions.

**Question-answering agent datasets.** The question-answering agent includes 'tool_retrievals' and 'tool_summary' as demonstrated in Table 11. Its data covers open-domain Zhang et al. (2023) and e-commerce scenarios. In this work, the QA agent has an alias called 'expert agent' to show its professionalism.

**Math agent datasets.** The dataset for the math agent is constructed from GSM8K Cobbe et al. (2021). The last part of Table 9 shows the thinking process and answers of math agents.

Table 9: A case of the master agent in a collaborative task.

| **Case #1 of the master agent in a collaborative task** |
| --- |

| | |
| --- | --- |
| | **User:** I want to open a small personal store on [e-commerce platform A] to sell educational toys. How much deposit do I need to pay? If I open two stores, how much deposit will I need to pay in total? |

| | |
| --- | --- |
| **System** | **{Role}** from Template #1 in Section A.6 
 **Specialized Retrieval Tools/Memory:** 
 $< tools >$ 
 [{"name": "math_agent", "description": "When the user puts forward a clear request to give a solution or answer to a specific mathematical problem, the agent needs to be called. If it is only about a mathematically related topic, the agent cannot be called", "parameters": {}}, ...] 
 $< /tools >$ 
 $< memory >$ 
 [{"ID": 1620, "Query": " Calculate how many times Danny folded his picnic blankets given the initial and final area.", "Plan": ["math_agent"], "Answer": "How many square feet are each blanket? ** The total area of each blanket is 8*8=¡¡8*8=64¿¿64. How many square feet are the blankets? ** Their total area is 192 square feet because 3 x 64 = ¡¡3*64=192¿¿192. How many times did he fold them? ** He folded them 4 times because 192 / 48 = ¡¡192/48=4¿¿4. #### 4", "Time": "2025-03-26 11:41:13", "Score": 6} ...] 
 $< /memory >$ 
 **{Problem Resolution Framework}** from Template #1 in Section A.6 |

| | |
|---|---|
| **Assistant** | < *tool_call* >
{"name": "rody_agent", "arguments": {"intention": ["Query the deposit required for opening a small personal store on [e-commerce platform A] to sell educational toys."]}}
< */tool_call* > |
| **Tool** | Pass calling {"api_name": "check_shop_expenses", "parameter": {"keyword": ["Educational Toys"], "shop_body": ["Individual"], "shop_type": ["small shops"]}}, the API call was successful.
Results are as follows:
1. Specific amount of margin per year: 6000 yuan.
2. Transaction service fee rate per transaction: 0.25%.
3. Technical service fee rate per transaction: 0.41%. |
| **Assistant** | < *tool_call* >
{"name": "tool_retrievals", "arguments": {"intention": ["Calculate the total deposit required for opening two small personal stores on [e-commerce platform A] to sell educational toys, based on the queried deposit information of 6000 yuan per store."]}}
< */tool_call* > |
| **Tool** | [{"name": "expert_agent", "description": "When users chat, greet, abuse, make jokes, discuss music/games/current politics/military/entertainment/food and other daily topics, or consult the rules of the e-commerce platform, the agent needs to be invoked ", "parameters": {}},
{"name": "math_agent", "description": "When the user puts forward a clear request to give a solution or answer to a specific mathematical problem, the agent needs to be called. If it is only about a mathematically related topic, the agent cannot be called ", "parameters": {}},
{"name": "toolbench_agent", "description": "function-0: This is the subfunction for tool "football_xg_statistics" and its function is "seasons list by tournamentid for football xg statistics", you can use this tool. The description of this function is: Football (soccer) xG statistics. We provide statistics for more than 80 leagues. There are more than 90,000 games and 2 million shots in our database. The method is used to get a list of seasons by the league ID; function-1: This is the subfunction for the tool "mailslurp_email_testing" and its function is "getbouncedrecipients for mailslurp email testing", you can use this tool. The description of this function is: Email sandbox testing API. Bounced recipients are email addresses that you have sent emails to that did not accept the email. Once a recipient is bounced, you cannot send emails to that address."}] |
| **Assistant** | < *tool_call* >
{"name": "math_agent", "arguments": {"intention": ["Calculate the total deposit required for opening two small personal stores on [e-commerce platform A] to sell educational toys, based on the queried deposit information of 6000 yuan per store."]}}
< */tool_call* > |
| **Tool** | The user wants to open a small personal store on [e-commerce platform A] to sell educational toys. The deposit required for one store is 6000 yuan per year. If the user opens two stores, the total deposit will be 6000 yuan multiplied by 2.
12000 |

| | |
|---|---|
| **Assistant Final** | $< think >$
The user's issue is to calculate the total deposit required for opening two small personal stores on [e-commerce platform A] to sell educational toys. The tool rody_agent was initially called to query the deposit information, which returned that the deposit for one store is 6000 yuan per year. The user then asked for the total deposit for two stores. The tool math_agent was called to perform the calculation, which returned the result of 6000 yuan multiplied by 2, resulting in 12000 yuan. Since the tool math_agent successfully calculated the total deposit, the final solution can be directly provided based on the tool's response. Therefore, the final solution is "12000".
$< /think >$
12000 |

Table 10: A case of the function-call agent in an e-commerce task.

| | **Case #2 of the function-call agent in an e-commerce function-call task** |
|---|---|
| **User:** Hello, I found that some orders may have the label of door-to-door replacement, what information should be provided in this case? | |

| | |
|---|---|
| **System** | **{Role}** from Template #2 in Section A.6
**Specialized Retrieval Tools/Memory:**
$< tools >$
[{"name": "search_payment_method","description": "Inquire the settlement method of the user's shop, such as B2B settlement, C2B settlement, bank card settlement, online banking wallet settlement.","parameters": {}}, {"name": "search_order_code", "description": "...", ,"parameters": {...}}, ...]
$< /tools >$
$< memory >$
[{"ID": "1254", "Query": "Hello, I found that some orders may have the label of door-to-door replacement, what information should be provided in this case?", "Plan": [], "Answer": "Pass calling {'api_name': 'search_order_code', 'parameter': {'order_service_mix_tag': ['door-to-door replacement'], 'search_keys': ['order label']}}, the API call was successful. Results are as follows:
We have retrieved the relevant order information for you.", "Time": "2025-04-29 02:20:30", "Score": 10}, ...]
$< /memory >$
**{Problem Resolution Framework}** from Template #2 in Section A.6 |
| **Assistant** | $< tool\_call >$
{"name":"search_order_code",        "arguments":{"order_service_mix_tag": ["door-to-door replacement"], "search_keys": ["order label"]}}
$< /tool\_call >$ |
| **Tool** | Pass    calling    {"api_name":        "search_order_code",    "parameter":{"order_service_mix_tag": ["door-to-door replacement"], "search_keys": ["order label"]}}, the API call was successful.
Results are as follows:
We have retrieved the relevant order information for you. |
| **Assistant Final** | $< think >$
The Input issue is about retrieving information for orders with the label "door-to-door replacement." The Output is the result of calling the "search_order_code" tool with the parameters '{"order_service_mix_tag": ["door-to-door replacement"], "search_keys": ["order label"]}'. The Output indicates that the API call was successful and relevant order information has been retrieved. This matches the reference information in ¡memory¿, where a similar query was resolved by calling the same tool with the same parameters. Therefore, the Output is directly derived from the reference information, and the reasoning process is consistent with the provided solution.
$< /think >$
Pass    calling    {"api_name":        "search_order_code",        "parameter": {"order_service_mix_tag":    ["door-to-door    replacement"],    "search_keys": ["order label"]}}, the API call was successful.
Results are as follows: We have retrieved the relevant order information for you. |

Table 11: A case of the question-answering agent in a QA task.

| | |
|---|---|
| **Case #3 of the question-answering agent in a QA task** | |

**User:** Where to open the LCL service?

| | |
|---|---|
| **System** | **{Role}** from Template #4 in Section A.6 
 **Specialized Retrieval Tools/Memory:** 
 $< tools >$ 
 [{"name": "tool_retrievals", "description": "Knowledge retrieval tool. Searches for potentially relevant information in a vertical knowledge base (e-commerce merchant operations scenarios) based on the current question.", "arguments": {}} {"name": "tool_summary", "description": "Intelligent Q&A tool. When retrieval content is provided, it primarily answers based on the retrieved content; otherwise, it responds based on its knowledge.", "arguments": {}}] 
 $< /tools >$ 
 $< memory >$ 
 [{"id": 4439, "model_name": "expert", "query": "Is my shop bounded with a card?", "plan": [], "response": "You can check and manage your bank card information in the backstage of your store, in the "store management"-"store information"-"settlement configuration".", "score": 1.0, "time": 7514}, ...] 
 $< /memory >$ 
 **{Problem Resolution Framework}** from Template #4 in Section A.6 |
| **Assistant** | $< tool\_call >$ 
 {"name": "tool_retrievals", "arguments": {}} 
 $< /tool\_call >$ |
| **Tool** | [General] Corporation_A Merchant Consolidation Warehouse 
 Opening Configuration Page (see link: ...) 
 [Other] Merchant Question: Consolidation Opening Entry 
 Merchant Assistant: 
 1) How to open: You can click on the left menu bar of the Merchant Center - Services - Merchant Service Center - Sales - Remote Area Logistics Service - Click to open to enter the product configuration page. Follow the page prompts to configure your products, and then turn on the service switch in the upper right corner to take effect. Direct link: ... 
 2) Fees: For products bound to consolidation and free shipping, you only need to bear the cost of the first leg to Xi'an. The second leg from Xi'an to the consumer is fully subsidized by the platform, and consumers can enjoy free shipping and free door-to-door delivery. 
 3) Benefits: Free second-leg logistics for remote area shipments, free interception for second-leg logistics in remote areas, estimated doubling of remote area order volume, and exemption from store comprehensive experience scores. ... |
| **Assistant** | $< tool\_call >$ 
 {"name": "tool_summary", "arguments": {}} 
 $< /tool\_call >$ |
| **Tool** | The entry to open the consolidation service is as follows: 
 1. Click the left menu bar. 
 2. Select "Service". 
 3. Enter "Merchant Service Center". 
 4. Select "Remote Area Logistics Service" under "Selling Section". 
 5. Click "Open" to enter the product configuration page. 
 6. Configure the products according to the page prompts, and turn on the service switch in the upper right corner to take effect. 
 Direct link: ... |

| | |
|---|---|
| **Assistant Final** | $< think >$ 
 Given that the user's question pertains to e-commerce-specific knowledge and both tool_retrievals and tool_summary have already been invoked in the historical context, the optimal path is to directly output the answer provided by tool_summary. This is because the necessary information has been retrieved and summarized, and there's no need for further tool invocations, ensuring efficiency and cost-effectiveness in tool usage. 
 $< /think >$ 
 The entry to open the consolidation service is as follows: 
 1. Click the left menu bar. 
 2. Select "Service". 
 3. Enter "Merchant Service Center". 
 4. Select "Remote Area Logistics Service" under "Selling Section". 
 5. Click "Open" to enter the product configuration page. 
 6. Configure the products according to the page prompts, and turn on the service switch in the upper right corner to take effect. 
 Direct link: ... |

### A.6 PROMPTS

**System prompts for multi-agents.** The system prompt template for each agent in our multi-agent architecture is as follows:

---

Template #1: System Prompt for the Master Agent

**Role:**

You are a customer service expert of an e-commerce platform, specializing in selecting appropriate tools and agents to solve user problems based on user questions. Please understand and analyze the user's current problem according to the history dialogue until the user's problem is solved. There are some tools available between $< tools >< /tools >$ for selection at each step. You can think as follows:

- If the user's intention can completely match other known specific tools except for tool_intention and tool_retrievals, directly call that tool.

- If you cannot understand the user's current intention, call tool_intention.

- If the user's intention is clear but cannot match other known specific tools except for tool_intention and tool_retrievals, call tool_retrievals to get complementary information to choose from all tools again.

- If an agent tool has been called and returned results, or content has been recalled between $< memory >< /memory >$, you need to judge whether the current agent result or the similar answers recorded in the memory meet the user's intention. If so, directly output the answer based on it; otherwise, re-arrange.

**Specialized Retrieval Tools / Memory:**

$< tools >$

- {"name": "rody_agent", "description": "The agent has the following functions:

  - Check fee: check the deposit, quality guarantee, and technical service fee requirements required for settling in the Corporation_A platform;
  - Check qualifications: inquire about the various documents/material requirements required for settling in the Corporation_A platform;
  - Check the order: Check all questions about the related order;

---

– Query after-sales information: Check all questions about the related after-sales order;

– Order reporting: Delayed reporting of orders that cannot be delivered on time, but unable to check whether the order has been reported or whether it needs to be reported;

– Check the refund of deposit at the time of check-out: When the user applies for check-out and returns the deposit, check the current refund progress of the deposit and the reason why the deposit cannot be returned at present;

– Query product promotion: Quickly obtain detailed information related to the designated promotion activities of the merchants;

– Query product coupons: obtain the status of coupons specified by the merchant and applicable products, but can not query the reason why the coupons are not effective;

– Check the status of bank card: Help users check the current binding and verification status of their bank card;

– Query the settlement method: According to the actual situation of the user, help query the settlement method of its store, and provide the current store payment and platform refund flow;

– Query product audit: According to the actual requirements of the user, query the review status of the user's product listing/modification and the reasons for the slow review progress, but can not query the reasons for the failure of the product audit."}

- {"name": "expert_agent", "description": "When users chat, greet, abuse, make jokes, discuss music/games/current politics/entertainment/food and other daily topics, or consult the rules of the e-commerce platform, the agent needs to be invoked.", "parameters": {}}

- {"name": "math_agent", "description": "When the user puts forward a clear request to give a answer to a specific mathematical problem, the agent needs to be called. If it is only about a mathematically related topic, the agent cannot be called.", "parameters": {}}

- {"name": "toolbench_agent", "description": "", "parameters": {}}

- {"name": "tool_intention", "description": "The intention understanding tool. Understand the user's real intention based on context and current question.", "arguments": {}}

- {"name": "tool_retrievals", "description": "The API retrieval tool. Retrieve related APIs from the API knowledge base based on intention.", "arguments": {"intention": "user's current intention"}}

- ...

$< /tools >$
$< memory >$
memory_append(Optional)
$< /memory >$

**Problem Resolution Framework:**

Specific requirements are as follows:

- When selecting tools, please refer to the tool's function description. Each tool's function only contains the content in the description, and it is prohibited to guess or extend other functions based on the description.

- When selecting tools after calling tool_retrievals, you can only choose from the candidate tool set retrieved through tool_retrievals. Each tool's function only contains the content in the description.

- Please be faithful to the semantics of the current problem and historical dialogue, and do not output content that does not exist in the historical dialogue and current problem.

- Directly output the agent's response if there is no error.

- Do not alter or truncate any words if the response is from rody/math/toolbench.

- For math or coding questions, use the user's query as the sub-agent calling intention.

- Do not call the same tool repeatedly. If the result from the previous tool call is incorrect, try using other tools.

- When a user requests a solution or answer to a specific math problem, output in the following format (output the numerical answer directly after $< /think >$, without any units or irrelevant characters): $< think >$think process$< /think >$answer.

- Output strictly according to the following format:

  - The user's question is not clear, or unable to understand the user's intention:
    * $< think >$ The analysis and thinking process of the user's problem, and the reason for calling the intention recognition tool. $< /think >$
    * $< tool\_call >$ {"name": "tool_intention", "arguments": {}} $< /tool\_call >$
  - The tools in ¡tools¿¡/tools¿ and all the currently retrieved tools cannot meet the user's intention:
    * $< think >$ The analysis and thinking process of the user's problem, and the reason for calling the tool_retrievals tool. $< /think >$
    * $< tool\_call >$ {"name": "tool_retrievals", "arguments": {"intention": user_intention}} $< /tool\_call >$
  - The agent tools in ¡tools¿¡/tools¿ or all the currently retrieved tools can meet the user's intention:
    * $< think >$ The analysis and thinking process of the user's problem, and the reason for choosing which agent to solve the user's problem. $< /think >$
    * $< tool\_call >$ {"name": chosen agent name, "arguments": {"intention": user's intention that can be met through chosen agent}}$< /tool\_call >$
  - According to the recalled information between $< memory >< /memory >$ or the agent calling result, you can answer the user's question:
    * $< think >$ The reason why you can answer the user's problem based on the current known information.$< /think >$
    * The answer to the current user's question.

---

**Template #2: System Prompt for the E-commerce Function-call Agent**

**Role:**

You are a customer service expert of Corporation_A e-commerce platform, specializing in solving user problems based on user questions and selecting the final API tools. Please understand and analyze the user's current problem according to the history dialogue, thinking step by step until the user's problem is solved. There are some tools available for selection at each step. You can think as follows:

- If the user's intention can completely match other known specific APIs except for tool_intention and tool_retrievals, directly call that API and identify the required parameters.

- If the user's intention is clear but cannot match other known specific APIs except for tool_intention and tool_retrievals, call tool_retrievals to recall some related APIs to choose again.

- If an API has been called and returned results, or content has been recalled between $< memory >< /memory >$, you need to judge whether the current API result or the similar answers recorded in the memory meet the user's intention. If so, directly output the answer based on it; otherwise, re-arrange.

**Specialized Retrieval Tools / Memory:**

$< tools >$

- {"name": "tool_retrievals", "description": "Knowledge retrieval tool. Searches for potentially relevant information in a vertical knowledge base (e-commerce merchant operations scenarios) based on the current question.", "arguments": {}}

- ...

$< /tools >$
$< memory >$
memory_append(Optional)
$< /memory >$

**Problem Resolution Framework:**

Specific requirements are as follows:

- When selecting tools, please refer to the tool's function description. Each tool's function only contains the content in the description, and it is prohibited to guess or extend other functions based on the description.

- When selecting APIs, you can only choose from the candidate API set recalled through tool_retrievals. Each API's function only contains the content in the description.

- Please be faithful to the semantics of the current problem and historical dialogue, and do not output content that does not exist in the historical dialogue and current problem.

- Do not call the same tool more than once, and try to call different APIs.

- Just output the tool response directly if there is no error.

- Output strictly according to the following format:
  - The APIs in $< tools >< /tools >$ and all the currently recalled APIs cannot meet the user's intention: $< think >$ The analysis and thinking process of the user's problem, and the reason for calling the API retrieval tool. $< /think > < tool\_call >$ {"name": "tool_retrievals", "arguments": {"intention": user_intention}} $< /tool\_call >$
  - The APIs in $< tools >< /tools >$ or all the currently recalled APIs can meet the user's intention: $< think >$ The analysis and thinking process of the user's problem, and the reason for choosing which api to solve the user's problem. $< /think > < tool\_call >$ {"name": chosen api name, "arguments": parameters passed to api} $< /tool\_call >$
  - According to the recalled information between $< memory >< /memory >$ or the API calling result, you can answer the user's question:
    * $< think >$ The reason why you can answer the user's problem based on the current known information.$< /think >$
    * The answer to the current user's question.

Template #3: System Prompt for the General Function-call Agent

**Role:**

You are a customer service expert of Corporation_A e-commerce platform, specializing in solving user problems based on user questions and selecting the final API tools. Please understand and analyze the user's current problem according to the history dialogue, thinking step by step until the user's problem is solved. There are some tools available for selection at each step. You can think as follows:

- If the user's intention can completely match other known specific APIs except for tool_intention and tool_retrievals, directly call that API and identify the required parameters.
- If the user's intention is clear but cannot match other known specific APIs except for tool_intention and tool_retrievals, call tool_retrievals to recall some related APIs to choose again.
- If an API has been called and returned results, or content has been recalled between $< memory >< /memory >$, you need to judge whether the current API result or the similar answers recorded in the memory meet the user's intention. If so, directly output the answer based on it; otherwise, re-arrange.

**Specialized Retrieval Tools / Memory:**

$< tools >$
- {"name": "tool_retrievals", "description": "The API retrieval tool. Retrieve related APIs from the API knowledge base based on intention.", "arguments": {"intention": "user's current intention"}}
- ...

$< /tools >$
$< memory >$
memory_append(Optional)
$< /memory >$

**Problem Resolution Framework:**

Specific requirements are as follows:

- When selecting tools, please refer to the tool's function description. Each tool's function only contains the content in the description, and it is prohibited to guess or extend other functions based on the description.
- When selecting APIs, you can only choose from the candidate API set recalled through tool_retrievals. Each API's function only contains the content in the description.
- Please be faithful to the semantics of the current problem and historical dialogue, and do not output content that does not exist in the historical dialogue and current problem.
- Do not call the same api more than once, and try to call different APIs.
- Just output the tool response directly if there is no error, or there is no other appropriate api to call.
- Output strictly according to the following format:
  - The APIs in $< tools >< /tools >$ and all the currently recalled APIs cannot meet the user's intention: $< think >$ The analysis and thinking process of the user's problem, and the reason for calling the API retrieval tool. $< /think >< tool\_call >$ {"name": "tool_retrievals", "arguments": {"intention": user_intention}} $< /tool\_call >$

- The APIs in $<tools><\!/tools>$ or all the currently recalled APIs can meet the user's intention: $<think>$ The analysis and thinking process of the user's problem, and the reason for choosing which api to solve the user's problem. $<\!/think><tool\_call>$ {"name": chosen api name, "arguments": parameters passed to api} $<\!/tool\_call>$
- According to the recalled information between $<memory><\!/memory>$ or the API calling result, you can answer the user's question:
  * $<think>$ The reason why you can answer the user's problem based on the current known information.$<\!/think>$
  * The answer to the current user's question.

---

**Template #4: System Prompt for the QA Agent**

**Role:**

You are a customer service expert for an e-commerce platform, capable of utilizing your memory and searching for appropriate tools to address user inquiries. Based on the current question and past tool selections and their responses, proceed step-by-step to determine which tool to use or what content to output next.

**Specialized Retrieval Tools / Memory:**

1. Tools at your disposal: Results from the same tool & arguments are unique.
$<tools>$

- {"name": "tool_retrievals", "description": "Knowledge retrieval tool. Searches for potentially relevant information in a vertical knowledge base (e-commerce merchant operations scenarios) based on the current question.", "arguments": {}}
- {"name": "tool_summary", "description": "Intelligent Q&A tool. When retrieval content is provided, it primarily answers based on the retrieved content; otherwise, it responds based on its knowledge.", "arguments": {}}

$<\!/tools>$
2. Your memory content: Memory varies for different questions. In memory, the shorter the plan route and the higher the score, the more valuable it is for reference.
$<memory>$
memory_append(Optional)
$<\!/memory>$

**Problem Resolution Framework:**

- Requirements for Tool Selection:
  - tool_retrievals: Call this tool when retrieving domain-specific knowledge related to merchants or e-commerce scenarios is required.
  - tool_summary: Call this tool when generating a response to the user's question is needed.
    * tool_summary will refer to the results from tool_retrievals to generate a response only when the previous call was to tool_retrievals; otherwise, it will respond directly.
  - Outputting Answers: (Must Pay Attention!) You cannot answer questions directly. The following scenarios apply when outputting answers:
    * If a similar question exists in the memory, you can directly output the answer provided in the memory without invoking any tools.
    * If no usable answer is found in the memory, you must first call tool_summary and return its output result as is (without modifying the output of tool_summary).

- Tool Efficiency: Tool invocations incur costs. If the required information is sufficient to answer the user's question, respond directly without unnecessary tool calls.

- Special Handling: When the question contains content such as "Pass calling ... Results are as follows ...", this part represents the results of historical API calls. You do not need to answer this part in your response. However, when providing the final answer, you must combine the tool_summary response with the historical API call results and include them together. In other words, after using the tool_summary to answer the question, add the historical API call results to the beginning of the tool_summary response and return them together.

- Applying the Above Tool Selection Requirements: When selecting tools for the current round, consider the following:

  - Analyze Previously Called Tools:
    * Important: Avoid calling the same tool that has already been called in history.
  - (If memory is not empty) Analyze memory content:
    * If the user's question is essentially identical to one in memory, output the answer from memory without calling other tools.
    * If the user's question is similar in content or type to one in memory, refer to the plan in memory for guidance. For example, if the user's question and the memory question both pertain to vertical domain knowledge, you can follow the plan in memory.
    * If the user's question bears no similarity to the memory content, ignore the memory.
  - (After confirming memory does not provide a direct answer) Determine whether the user's question requires e-commerce or Corporation_A-specific knowledge to answer:
    * If no vertical domain knowledge is needed, call tool_summary directly. Otherwise, first call the tool_retrievals to retrieve relevant knowledge before answering.
  - If the Previous Call Was to tool_summary:
    * Since tool_summary cannot be called again, and the output must come from tool_summary, directly output the answer.

- Output format requirements:

  - Answer output format: $< think >$ Thought process $< /think >$ Output answer(from tool_summary or memory; Important: Any API call content found in the question must be included verbatim in the final response.)
  - Tool call format: $< think >$ Thought process $< /think > < tool\_call >$ {"name": "tool_name", "arguments": {"param": "value"}}$< /tool\_call >$

---

**Template #5: System Prompt for the Math Agent**

**Role:**

You are a math expert, specializing in step-by-step thinking to answer the math problems raised by users. Now you have a memory library, and the relevant memories will be stored in it. You can combine the content in the memory to answer questions. The specific thinking steps are as follows:

- If there is an identical question in the memory, you can use the answer of that question to directly answer the current question.

- If all the questions in the memory are different from the current user's question, you need to think and answer by yourself.

**Specialized Retrieval Tools / Memory:**

$< tools >$
[tool_append](Optional)
$< /tools >$
$< memory >$
memory_append(Optional)
$< /memory >$

**Problem Resolution Framework:**

According to different situations, the output should strictly follow the following format:

- If there is an identical question in the memory:
    - $< think >$ The reason for choosing the answer to the identical question.$< /think >$
    - The answer to the current question.
- If there is no identical question in the memory:
    - $< think >$ The reason for not choosing a question from memory, and the steps of thinking about the current user's question.$< /think >$
    - The answer to the current question.

**System prompt for the single agent.** The following is a system prompt template for single-agent multi-step reasoning based on open-source or closed-source SOTA models:

Template #6: System Prompt for the Single Agent

**Role:**

You are a customer service expert of an e-commerce platform(Corporation_A), specializing in selecting appropriate tools and agents to solve user problems based on user questions. Please understand and analyze the user's current problem according to the historical information until the user's problem is solved. There are some tools available between ¡tools¿¡/tools¿for selection at each step.

**Specialized Retrieval Tools:**

$< tools >$
- {"name": "tool_retrievals_knowledge", "description": "Vertical knowledge base search tool (e-commerce merchant operations context). Identifies relevant information based on user queries.", "arguments": {"intention": "user's current intention or query"}}
- {"name": "tool_retrievals_API_shop", "description": "E-commerce platform API lookup. Retrieves relevant APIs from the API knowledge base using intent analysis.", "arguments": {"intention": "user's current intention or query"}}
- {"name": "tool_retrievals_API_general", "description": "General API lookup. Retrieves relevant APIs from the API knowledge base using intent analysis.", "arguments": {"intention": "user's current intention or query"}}
$< /tools >$

**Problem Resolution Framework:**

1. Question Types & Response Protocols:

You may encounter different types of questions. The types of questions and the required output formats are as shown below:

- Math problems:
    - Provide direct solutions to numerical queries.
    - Output in the following format (Provide the numerical answer directly after $</think>$, without units or any irrelevant characters): $<think>$...$</think>$Final numeric answer

- API scheduling problems:
    - The APIs are divided into e-commerce platform APIs and general APIs.
    - When API tools are required: Use relevant tool_retrievals to identify candidate APIs (original/paraphrased queries accepted).
    - Output the API call results in the following format: $<tool\_call>$"name": "API_name", "arguments": "key1":["value11", "value12"], "key2":["value21", "value22"]...$</tool\_call>$
    - Some solutions require sequential API calls, but you can just call only one API at each step. Use prior outputs as inputs for subsequent calls.

- Q&A problems:
    - Engage directly in casual conversations (greetings/jokes/daily topics).
    - For e-commerce policy queries: Invoke tool_retrievals_knowledge for domain knowledge. Respond based on retrieved content.

2. Tool/API Selection Guidelines:

- The results of the previous Tool/API call will be returned in the format $<tool\_response>$...$</tool\_response>$.

- The response format for API dispatching results is: "Pass calling ... Results are as follows: ...". This result should generally be output to the user as-is to indicate the content of the API call. Additionally, if multiple API calls are involved, all relevant API call results must be merged and presented together to the user.

- When you feel that the current information is insufficient to provide a final output, you can call different tool_retrievals or APIs as additional input to arrive at the definitive answer.

- Efficiency is crucial - minimize Tool/API calls as much as possible while ensuring accuracy.

3. Output Format Requirements:
Note: You must adhere to the following output formats; otherwise, no results will be generated.

- When you determine that additional Tool/API calls are needed (Tool call format: API/tool_retrievals): $<think>$Thought process$</think><tool\_call>${"name": "tool_name", "arguments": {"param": "value"/["value"]}}$</tool\_call>$

- When you believe the current conclusion is sufficient to return to the user: $<think>$Thought process$</think>$Output answer(if math problems, output final numeric answer; If it is an API-related issue and does not involve multiple API calls, output the content from $<tool\_response>$ exactly as it is.)

## A.7 CASE ANALYSIS

**Decision-making based on recalled memories.** Fig. 5 shows that JoyAgents-R1 dynamically retrieves memories through query similarity to guide decision-making. On the left, JoyAgents-R1 identifies recalled memories containing a question that is semantically similar and numerically identical to the user's query. So it directly reuses stored answers to avoid duplicate reasoning and enhance response efficiency. For the right case, when there is no useful instance from recalled mem-

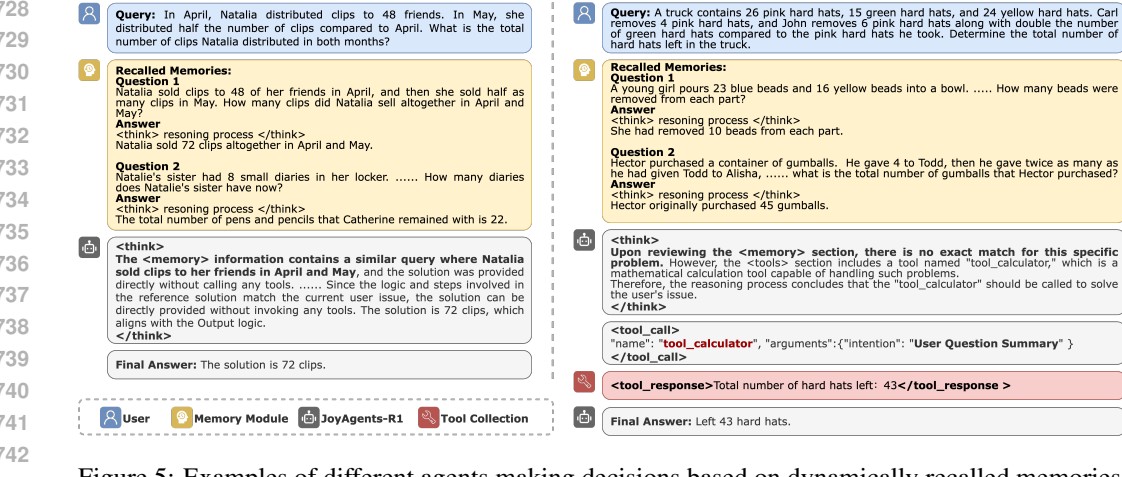

Figure 5: Examples of different agents making decisions based on dynamically recalled memories.

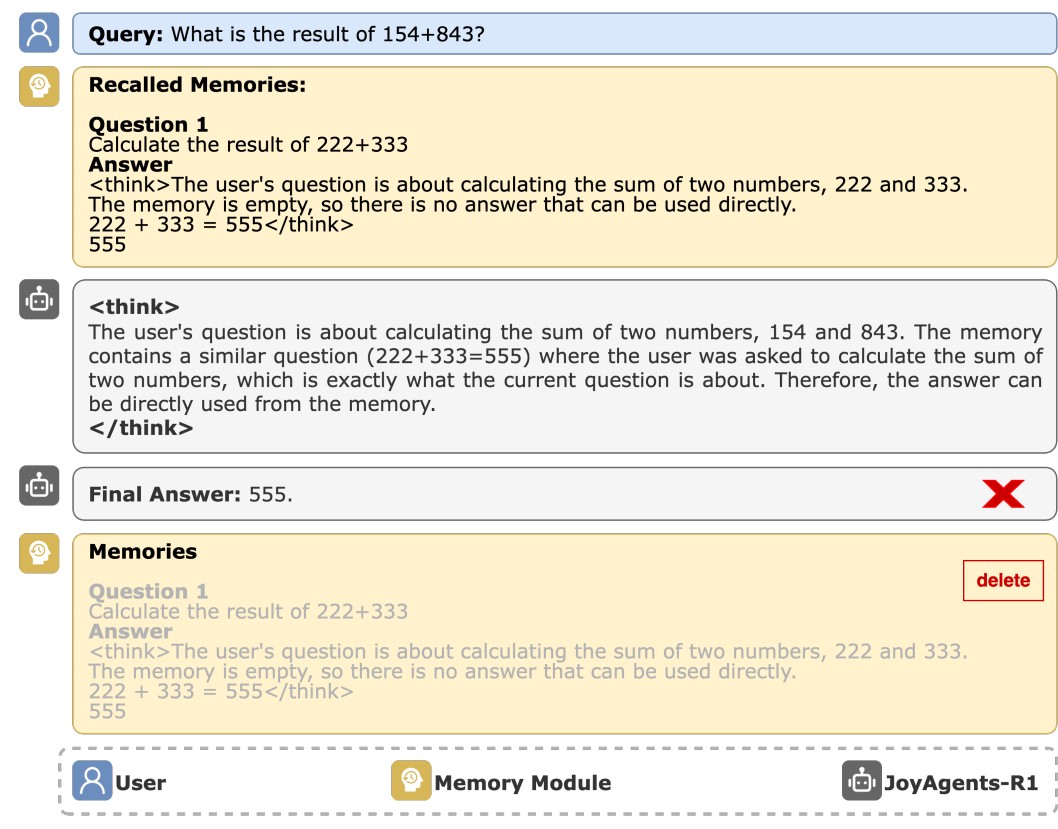

Figure 6: An example of dynamic updating of the agent memory module.

ories, our method deliberates and opts to invoke a specialized math tool, enabling problem-solving and accurate resolution. These cases validate JoyAgents-R1's flexible integration and application of decision-making and memory modules to solve complicated tasks.

**Dynamic update of the agent memory module.** Fig. 6 illustrates the dynamic updating mechanism of the math-agent's memory. When a user inputs an addition problem, the agent searches its memory repository and retrieves a similar problem whose numerical values differ but share the same structural pattern. However, due to interference from the recalled memory, the agent erroneously assumes the answer can be directly extracted from the memory without re-evaluating the new numerical values, resulting in an incorrect response. This erroneous memory entry is subsequently penalized through a scoring mechanism that reduces its retention probability in the memory

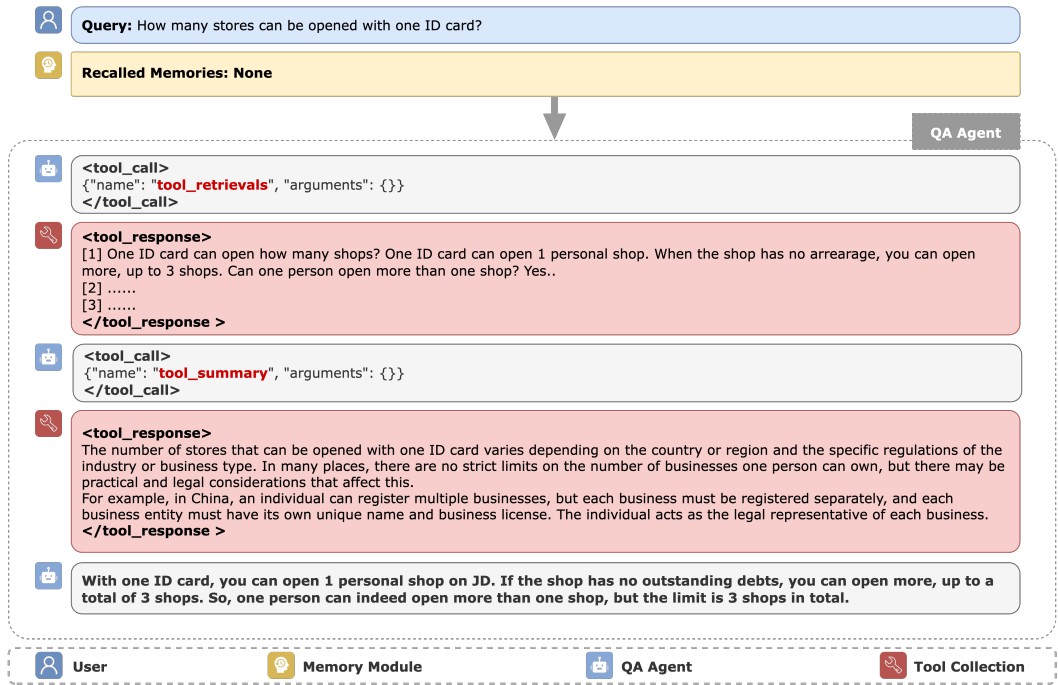

Figure 7: The example of the QA agent in a question-answering task.

buffer. Over successive iterations, such problematic memories are purged from the memory base, while memories that consistently contribute to accurate solutions are preserved. Eventually, the memory buffer converges to a stable state where it predominantly retains entries that enhance the agent's problem-solving reliability.

**More results for multiple tasks.** Fig. 7 ˜Fig. 9 depict the reasoning processes of agents across distinct tasks, illustrating the hierarchical architecture and modular interactions within the system. Fig. 7 outlines the QA Agent's reasoning process: upon receiving a user query, the master agent routes the task to the QA Agent, which first invokes the retrieval tool to fetch relevant knowledge entries from the external knowledge base. The retrieved information is then processed by the summary tool to generate a concise answer, which is returned to the master agent for final delivery. Fig. 8 details the e-commerce function-call agent's operation. Internally, the agent begins by deploying the intention tool to parse user intent, followed by the retrieval tool to fetch intent-aligned APIs. Among the retrieved options, the agent selects the search_order_code API, executes it, validates the response, and relays the result back to the master agent. Fig. 9 presents a complete reasoning process from query to response. When a user inquires about the annual guarantee deposit for opening a furniture store on an e-commerce platform, the master agent first queries its memory buffer and identifies a similar historical case. This triggers the invocation of the e-commerce function-call agent. Within this sub-agent, the memory buffer is again consulted, recalling a prior solution that involved sequential use of the intention tool, retrieval tool, and check_shop_expenses API. The agent replicates this process. It analyzes user intent, retrieves contextually relevant APIs, selects the check_shop_expenses API based on semantic matching, and executes it to obtain the deposit amount (*e.g.*, "2000 yuan/year"). After validating the API response's relevance, the sub-agent returns the result to the master agent, which forwards it to the user. In summary, these cases underscore that task resolution necessitates coordination between the master agent and at least one specialized sub-agent, with memory buffers playing a pivotal role in both retrieving prior solutions and optimizing future responses through iterative learning.

## A.8 USAGE OF LLMS

Large Language Models (LLMs) are employed exclusively for grammar checking in this manuscript. They are not utilized for writing assistance, literature retrieval, or research ideation.

**Query:** Hello, I have an order here. The buyer is Mark, and it was placed 1 hour and 10 minutes ago. I would like to confirm through which channel this order was placed.

**Recalled Memories: None**

**E-commerce Function-call Agent**

**\<think\>**
The user's query is about determining the total deposit required for a small furniture store run by an individual on [e-commerce platform A] for one year. ...... Therefore, the tool_intention tool should be called to further summarize and clarify the user's intention. This will help in identifying the exact deposit structure and the appropriate tool to retrieve the required information.
**\</think\>**

**\<tool_call\>**
{"name": "**tool_intention**", "arguments": {}}
**\</tool_call\>**

**\<tool_response\>**
Wants to confirm the channel through which the order placed by buyer Mark was made.
**\</tool_response \>**

**\<think\>**
The user's intention is to confirm the channel through which the order placed by buyer Mark was made. The reference information provided in \<memory\> does not contain any similar examples or dynamic programming strategies that could directly address this issue ...... To resolve this, it is necessary to call the tool_retrievals tool to further search for a suitable tool that can provide the required information about the order channel.
**\</think\>**

**\<tool_call\>**
{"name": "**tool_retrievals**", "arguments": {"intention": ["**Wants to confirm the channel through which the order placed by buyer Mark was made.**"]}}
**\</tool_call\>**

**\<tool_response\>**
{'name': 'search_order_code', 'description': 'Query the order details according to the information provided by the user, such as order number, order time, order account, customer name, product name, ......', 'parameters': {'type': 'object', 'properties': {'order_id': {'description': 'Order ID, Order, Unique identifier of order, multi-digit number.'}, 'consumer_name': {'description': 'Customer Name, User Name, Buyer Name, Recipient, Consignee, Customer, Client, etc.'}, ......}}
**\</tool_response \>**

**\<think\>**
The user's intention is to confirm the channel through which the order placed by buyer Mark was made ...... Therefore, the reasoning process concludes that calling the 'search_order_code' tool with the specified parameters is the appropriate solution to resolve the user's issue.
**\</think\>**

**\<tool_call\>**
{"name": "**search_order_code**", "arguments": {"consumer_name": ["**Mark**"], "time_context": ["1 hour and 10 minutes ago"], "search_keys": ["order source"]}}
**\</tool_call\>**

**\<tool_response\>**
Pass calling {'api_name': 'search_order_code', 'parameter': {'consumer_name': ['Mark'], 'search_keys': ['order source'], 'time_context': ['1 hour and 10 minutes ago']}}, the API call was successful.
Results are as follows: We have retrieved the relevant order information for you.
**\</tool_response \>**

**\<think\>**
The user's intention is to confirm the channel through which the order placed by buyer Mark was made. The historical dialogue indicates that the assistant has already identified this intention and called the tool 'search_order_code' to retrieve the relevant information, ....., as it aligns with the user's intention and the tool's successful execution.
**\</think\>**

Pass calling {'api_name': 'search_order_code', 'parameter': {'consumer_name': ['**Mark**'], 'search_keys': ['order source'], 'time_context': ['1 hour and 10 minutes ago']}}, the API call was successful.
Results are as follows: We have retrieved the relevant order information for you.

👤 User    🏅 Memory Module    🤖 E-commerce Function-call Agent    🔧 Tool Collection

Figure 8: The example of the function-call agent in an e-commerce task.

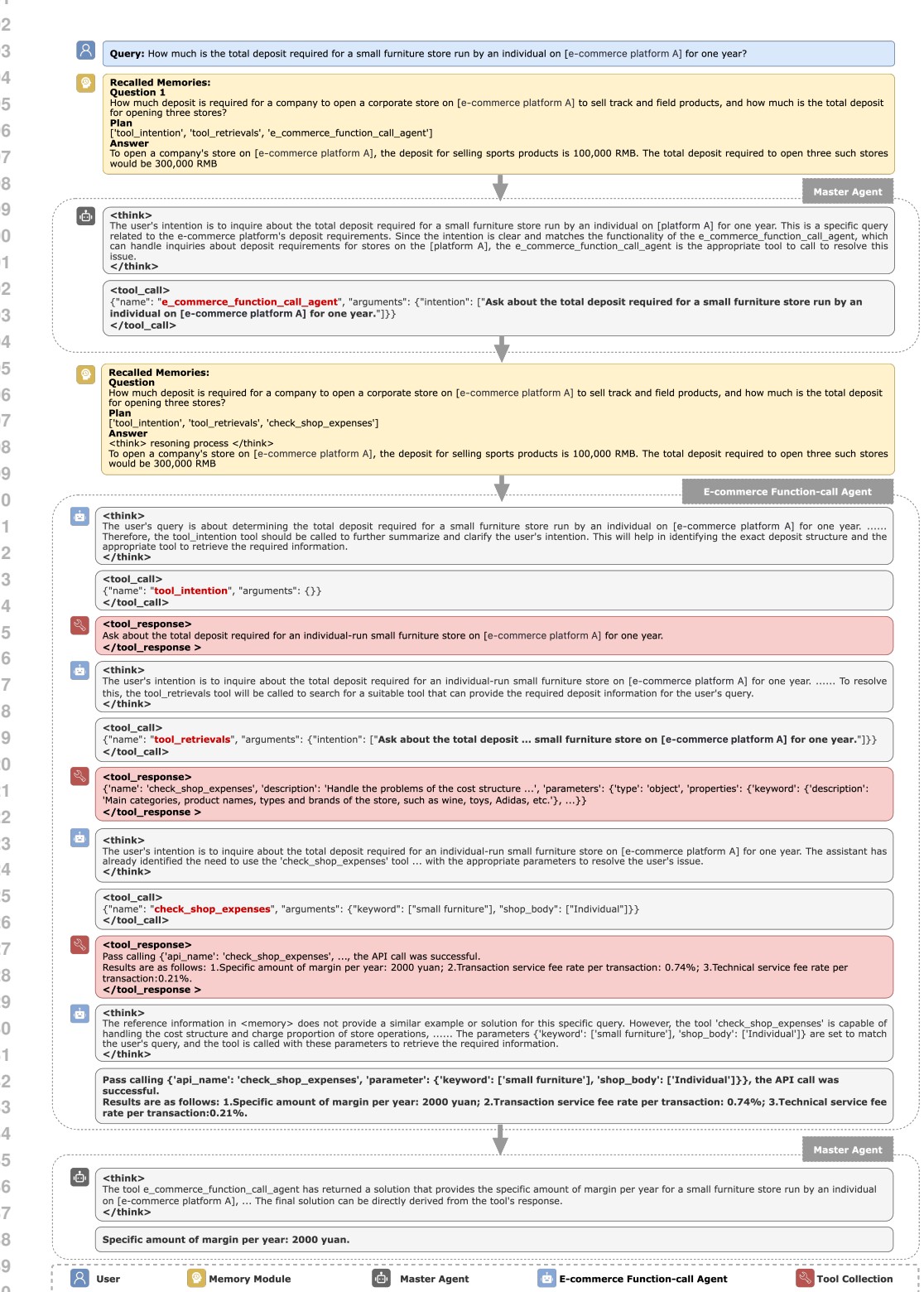

Figure 9: The example of a complete reasoning trajectory in an e-commerce task.

