# OpenReview forum: "JoyAgents-R1: Accelerating Multi-Agent Evolution Dynamics with Variance-Reduction Group Relative Policy Optimization"
_ICLR.cc/2026/Conference — ICLR 2026 Conference Withdrawn Submission_

### Official Review · Reviewer_erZT · 2025-10-18

**Soundness:** 2
**Presentation:** 3
**Contribution:** 3
**Rating:** 4
**Confidence:** 4

**Summary:**

This paper introduces JoyAgents-R1, a framework that accelerates multi-agent evolution with a novel Variance-Reduction Group Relative Policy Optimization (VR-GRPO), integrating efficient sampling and update strategies. Specifically, VR-GRPO performs Monte Carlo sampling based on an initial reasoning trajectory to avoid the exponential explosion of the joint action space while maintaining policy diversity. Then, the method selects the top-K sampling groups with maximal reward fluctuations based on the marginal benefit principle, thereby enabling cost-effective parameter updates. To further complement evolution, an adaptive memory evolution mechanism that repurposes GRPO rewards as cost-free supervisory signals is designed to eliminate repetitive reasoning and accelerate convergence.

**Strengths:**

1, The paper claims that it is the first work to adapt GRPO for functionally distinct multi-agents, enabling synergistic enhancement of their decision-making and memory capabilities.
2, The paper proposes Variance-Reduction Group Relative Policy Optimization (VR-GRPO), tailored for multi-agent systems.

**Weaknesses:**

I think the main weakness is in the experiment part. Please see details in the questions.

**Questions:**

1, In A.4 MORE ANALYSIS FOR ABLATION STUDY, for title "Ablation on updating top-K models" you mean nodes here?
2, How much nodes in total? I see the table 4 compares top1, 2, 5, all. But if the total nodes are like 20 or even more than a hundreds, can you maybe show like top 4,6,8,10? And also I am curious about how general top-5 is, like if using different model in different datasets, would top-5 always be the best or you would actually need to carefully tune the parameter k?
3, For the design of the main table, is there any related articles I could follow or refer to? Like how previous works did the comparison between their proposed method and baselines? And why you choose these as your baselines? Or add a simple explanation of your selected baselines.
4, For the closed-source models used in the paper, could you show maybe chatgpt 5 and Claude4.0-sonnet?
5, In table 1, I could tell that for JoyAgents-R1 larger Master leads to a better Avg performance. But I am wondering what would the performance be if use like 3B Master + 7B Sub agents? And also could you scale up the master to 14B to test the scalability of your method? For example would JoyAgents-R1(14B Master + 3B Sub agents) directly outperform the JoyAgents-R1(7B Master + 3B Sub agents)? This concern is due to some specific tasks, it seems like smaller Master performs better than larger Master.
6, For all the experiments and evaluation of proposed method, Qwen2.5 is the base model family. I think it would be necessary to show the results from at least one more model family such as Llama3, Llama2 or any other open source model families.
7, Did you run your experiments across different random seeds?
8, Could you also provide the comparison results for JoyAgents-R1(7B Master + 3B Sub agents), JoyAgents-SFT (3B+3B) and JoyAgents-SFT-no (3B+3B) in table 2?

---

### Official Review · Reviewer_7gUC · 2025-10-27

**Soundness:** 3
**Presentation:** 3
**Contribution:** 3
**Rating:** 6
**Confidence:** 4

**Summary:**

The paper introduces JoyAgents-R1, a new framework for Multi-Agent Reinforcement Learning (MARL) designed to address cooperative inefficiency and training instability in heterogeneous agent systems. The approach integrates a hierarchical architecture comprising a master agent and specialized sub-agents (e.g., Q&A, Function-call, Math). The core of the method is a variance-reduction GRPO tailored for MARL, which includes node-wise monte carlo sampling to manage trajectory explosion and a marginal benefit-driven selection strategy for efficient parameter updates. Furthermore, JoyAgents-R1 incorporates an adaptive memory evolution mechanism that leverages GRPO rewards as a "cost-free supervisory signal" to improve convergence and eliminate repetitive reasoning. The overall goal is to achieve performance comparable to larger LLMs using smaller, open-source models.

**Strengths:**

1. The paper clearly identifies and proposes solutions for key challenges in applying traditional policy optimization methods like GRPO to heterogeneous MARL: **low sampling efficiency** and **slow convergence**. This shows a strong grasp of the practical difficulties in LLM-based MARL.
2. Novel variance-reduction GRPO adaptation. In this method, node-wise monte carlo sampling greatly improves sampling efficiency, while marginal benefit-diven model update addresses training stability and efficiency.
3. Cost-effective memory evolution using the $R_A+R_F$ rewards as a cost-free supervisory signal for memory evolution is highly appealing. This "free lunch" approach avoids the need for a separate memory training loss or a dedicated, potentially large, supervisory model, which is excellent for cost-effectiveness and accelerating convergence. The dynamic memory updating algorithm provides a concrete mechanism for memory curation based on reward thresholds.

**Weaknesses:**

1. Clarity the marginal benefit principle. The connection between reward variance and the marginal benefit principle requires stronger theoretical justification. While high variance intuitively indicates high potential for improvement (or deterioration), the paper does not **formally prove** or give empirical experiments that there is a phenomenon of excessive variance in MARL, and if the method maximizes $Var(R_i )$ for parameter updates the marginal joint benefit for the overall multi-agent system.
2. Maybe potential bias in reward definition? the reward $R_E$ is excluded from the memory update reward but included in the policy optimization reward ($R=R_A+R_F-R_E$). This inconsistency can lead to a divergence between the memory's preference (optimized for accuracy/format) and the agent's policy preference (optimized for accuracy/format/efficiency).
3. How to effectively allocate rewards to each agent, is this a CTDE framework? Agents are heterogeneous, why can they be jointly optimized in this way?
4. Limited heterogeneity and scalability demonstration: The current architecture features a Master and four specific sub-agents (QA, EFC, GFC, Math). While heterogeneous in function, the underlying LLM backbone is the same. The claim of handling heterogeneous agents would be stronger if the framework were tested with agents based on fundamentally different architectures or a much larger number of sub-agents to truly test the limits of the VR-GRPO's scalability.
5. Algorithm 1 uses a reward thresholding mechanism based on an approximate normal distribution (2.5% and 97.5% percentiles) for memory updates. A brief justification for the normality assumption or a comparison with a simpler fixed threshold approach (which would be an interesting ablation) is missing. The time decay and reward difference terms in line 15 also lack a clear justification for setting α=β=1.
6. Perhaps it is necessary to add a line of performance for untrained JoyAgents.

**Questions:**

See above.

---

### Official Review · Reviewer_2MVW · 2025-10-31

**Soundness:** 2
**Presentation:** 3
**Contribution:** 2
**Rating:** 2
**Confidence:** 4

**Summary:**

This paper focuses on variance control in the training process of the Group Relative Policy Optimization (GRPO) algorithm within LLM-based multi-agent systems. The overall training framework is similar to the decentralized training approach in Multi-Agent Reinforcement Learning (MARL), where each agent optimizes its policy assuming the policies of other agents remain fixed, solely using GRPO to optimize its own policy. Through this approach, the proposed algorithm intuitively achieves higher exploration efficiency compared to directly exploring the joint action space, potentially reducing the exponential complexity of action space exploration to a linear level. Furthermore, the authors introduce a refinement in the policy optimization phase, which is the main step for their variance control method: by sorting the variance of the grouped rewards during each agent's policy learning, they determine which agent policies to optimize in the current iteration. The experiments are conducted using Qwen2.5 as the base model, tested on multiple SFT and RL tasks. The tested baselines include mainstream open-source and non-open-source models/algorithms. The comparative results show that the average performance is comparable to that of DeepSeek-R1.

**Strengths:**

- This paper is easy to follow
- The proposed method shows as a plugin-in which can be implemented compatibly with existing methods.

**Weaknesses:**

- The key components of the algorithm lack strong experimental verification, with only a few experiments provided to support them. This includes: 1) whether the proposed variance reduction method impacts the convergence of the algorithm; 2)the Monte Carlo sampling is similar to a coordinate descent method, and its limitations and scope of applicability are not discussed.
- The designed experiments fail to highlight the advantage of the proposed algorithm in terms of variance control and exploration efficiency improvement.

**Questions:**

- What exactly do M1-M6 in Table 3 represent? I suggest providing a brief explanation in the caption, as the current layout is not clear.
- Why are there no learning curves provided?
- JoyAgents-R1 (7B Master + 3B Sub agents) uses a larger model but exhibits poorer performance. Why is this the case?

---

### Official Review · Reviewer_PoeU · 2025-11-01

**Soundness:** 3
**Presentation:** 3
**Contribution:** 2
**Rating:** 6
**Confidence:** 4

**Summary:**

This paper introduces JoyAgents-R1, a novel framework for accelerating multi-agent reinforcement learning through Variance-Reduction Group Relative Policy Optimization (VR-GRPO). The method addresses the computational challenges of joint optimization across heterogeneous agents by employing Monte Carlo sampling along initial trajectories to avoid exponential action space explosion, selecting top-K agents with maximal reward variance for efficient updates, and incorporating an adaptive memory evolution mechanism that leverages GRPO rewards as supervisory signals. Experimental results on multi-task AI assistant benchmarks demonstrate that the framework, built on smaller 3B/7B models, achieves performance comparable to much larger models like DeepSeek-R1.

**Strengths:**

1. The paper presents the application of GRPO to functionally distinct multi-agents, introducing VR-GRPO with efficient Monte Carlo sampling and marginal benefit-driven updates that effectively manage the exponential growth of joint action spaces.
2. The framework cleverly repurposes GRPO rewards as "free lunch" supervisory signals for memory updates, enabling synchronous optimization of decision-making and memory modules without requiring additional training overhead.
3. Despite using significantly smaller models (3B/7B parameters), JoyAgents-R1 achieves competitive results against much larger systems, demonstrating excellent parameter efficiency and practical applicability for resource-constrained deployments.

**Weaknesses:**

1. The computation and utilization of similarity metrics in Algorithm 1 lack clear specification, making it difficult to understand how similarity scores influence memory updates and agent decision-making processes.
2. The experimental datasets primarily consist of isolated tasks, and the paper does not adequately explain how the 100 collaborative task instances were constructed, raising questions about the framework's evaluation on truly interdependent multi-agent scenarios.
3. The observation that 3B Master + 3B Sub agents outperforms 7B Master + 3B Sub agents on EFC and GFC tasks suggests potential scalability issues, indicating the framework may be more dependent on sub-agent improvements rather than master agent capacity.
4. Figure 4 shows that removing memory initially improves performance before declining, which contradicts the expected monotonic degradation and raises questions about the true contribution of the memory mechanism to overall system performance.

**Questions:**

1. How does the framework handle more complex real-world scenarios beyond the current agent types (e.g., file search, application control), and what modifications would be necessary to extend the architecture?
2. What is the computational overhead of the VR-GRPO sampling strategy compared to standard GRPO, particularly as the number of agents and trajectory length increase?
3. Could the authors provide more analysis on why the memory-free variant shows initial performance improvements, and how this relates to the GRPO training dynamics?
4. Given the hierarchical architecture, how does error propagation from the master agent affect sub-agent performance, and are there mechanisms to mitigate cascading failures?
5. What are the convergence guarantees of VR-GRPO compared to standard GRPO, particularly when only updating a subset of agents based on variance?

---

### Official Review · Reviewer_VQxJ · 2025-11-03

**Soundness:** 2
**Presentation:** 2
**Contribution:** 2
**Rating:** 2
**Confidence:** 4

**Summary:**

Summary:
The paper proposes JoyAgents-R1, a multi-agent reinforcement learning framework based on Variance-Reduction Group Relative Policy Optimization (VR-GRPO). It claims to improve multi-agent coordination and efficiency through (1) Monte Carlo trajectory sampling, (2) variance-based top-K updates, and (3) memory evolution.

**Strengths:**

The system architecture (Master/Sub-agents) and training pipeline (SFT → RL fine-tuning) are clearly described.
Experiments is solid, including multi-domain evaluations (general reasoning, e-commerce, function calling).

**Weaknesses:**

1. Lack of Novelty
The proposed VR-GRPO method is not substantively new.
The combination of Monte Carlo sampling + selective update based on variance has already been explored in several existing agentic RL and multi-agent PPO frameworks.
Specifically, the paper’s “trajectory-based Monte Carlo sampling” is nearly identical to prior works on *Monte-Carlo Tree Search in Multi-Agent RL* and *Monte-Carlo Sampling for Agentic LLM RL*.
In addition, the “variance reduction + top-K update” heuristic is a mild optimization tweak, not a new learning principle.

2. Incomplete Comparison
JoyAgents uses multi-agent specialization, while baselines are single-model. The paper only compares against LLM models (DeepSeek-V3, DeepSeek-R1, GPT-4o), rather than other agentic RL methods.
It completely omits comparisons with recent multi-agent RL frameworks, such as MLPO mentioned in related work.

**Questions:**

Why EFC for most of the llm model is extremely low, even for gpt-4o-mini and Claude3.5-sonnet?
What is “Cooperation” task mentioned in Table 1? The paper only said “The test set … 100 cases for the collaborative task”.

---

### Note · Authors · 2026-01-06

I have read and agree with the venue's withdrawal policy on behalf of myself and my co-authors.